# Towards Repeatable, Scalable Graphene Integrated Micro-Nano Electromechanical Systems (MEMS/NEMS)

**DOI:** 10.3390/mi13010027

**Published:** 2021-12-26

**Authors:** Joon Hyong Cho, David Cayll, Dipankar Behera, Michael Cullinan

**Affiliations:** Department of Mechanical Engineering, The University of Texas at Austin, 204 E Dean Keeton St, Austin, TX 78712, USA; joonyboy@utexas.edu (J.H.C.); dcayll@utexas.edu (D.C.); dipankar.behera@utexas.edu (D.B.)

**Keywords:** graphene, chemical vapor deposition (CVD), MEMS, NEMS, sensor, resonator, repeatability, scalability

## Abstract

The demand for graphene-based devices is rapidly growing but there are significant challenges for developing scalable and repeatable processes for the manufacturing of graphene devices. Basic research on understanding and controlling growth mechanisms have recently enabled various mass production approaches over the past decade. However, the integration of graphene with Micro-Nano Electromechanical Systems (MEMS/NEMS) has been especially challenging due to performance sensitivities of these systems to the production process. Therefore, ability to produce graphene-based devices on a large scale with high repeatability is still a major barrier to the commercialization of graphene. In this review article, we discuss the merits of integrating graphene into Micro-Nano Electromechanical Systems, current approaches for the mass production of graphene integrated devices, and propose solutions to overcome current manufacturing limits for the scalable and repeatable production of integrated graphene-based devices.

## 1. Introduction

For over a decade graphene has been studied as a 2D material with numerous favorable properties including high strength and elastic modulus [1], high electron mobility [2], and high thermal conductivity [3]. However, the 2D nature of graphene makes it difficult to manufacture and integrate for commercial applications in sensors, switches, and other MEMS/NEMS devices. While large scale manufacturing is possible [4], studies show that material properties can change depending on the process [5]. For manufacturing graphene at a large scale, it is important to understand how the parameters affecting production speed and the quality of graphene are related. It is generally known that the growth of graphene at higher temperature produces higher quality graphene [6]. There is also significant ongoing research dedicated to finding out important growth parameters related to increasing the production scale, decreasing growth time, and increasing the quality of graphene. This research is key for graphene to become viable for integrated applications in MEMS/NEMS, especially in state-of-the-art devices that aim to effectively utilize graphene’s extraordinary properties.

There are obstacles for graphene to be implemented in MEMS/NEMS devices because the mass production of graphene has repeatability and scalability issues [7]. For example, graphene-based MEMS/NEMS resonators can exhibit significantly different resonant frequencies even if they are fabricated in the same batch with the same process [8]. In addition, many graphene-based MEMS/NEMS devices require suspended graphene structures for the device to benefit from exceptional electrical and mechanical properties of graphene. However, suspending graphene is challenging because the transfer and release process often introduces defects into the devices which researchers need to address in addition to intrinsic defects already present in the graphene.

In this review, we will introduce several on-going research efforts focusing on increasing graphene production scale, speed, and quality. We will also highlight the current challenges for graphene to be successfully implemented in MEMS/NEMS device applications at a large scale and suggest how these difficulties can be overcome.

## 2. Why Graphene for MEMS/NEMS

Graphene is an atomically thin 2D carbon material with an sp^2^ bonded honeycomb lattice structure. The theoretical foundations of graphene (or ‘2D graphite’) were introduced by P.R. Wallace who sought to calculate the band structure and the physical characteristics of monocrystalline graphite [9]. However, the existence of stable graphene was difficult to prove. The most conclusive evidence of the stability and existence of graphene was presented by Novoselov et al. [10]. They were able to successfully isolate graphene from a bulk graphite crystal using a mechanical exfoliation technique. Pressure-sensitive tape was used to remove the top few layers of the graphite crystal and it was then transferred onto a Si/SiO_2_ substrate. This process was repeated on the transferred films until only a single monolayer of graphene was exfoliated. In addition to the ability to isolate graphene in such an inexpensive manner, this work highlighted the impressive electronic properties of graphene. Andre Geim and Konstantin Novoselov were awarded the Nobel Prize in 2010 for the discovery of graphene and its groundbreaking properties.

## 3. Graphene Properties

Since its discovery in a freestanding form in 2004, graphene has surpassed the expectations of researchers around the globe. Graphene has remarkable mechanical, electrical, magnetic, thermal, and optical properties. There are several review articles that discuss these properties in detail [11,12,13,14]. The following sections briefly discuss the key mechanical, electronic, thermal, and electromechanical properties of graphene with a focus on MEMS/NEMS-based applications.

### 3.1. Mechanical Properties

The excellent mechanical properties of graphene can be attributed to the strength of the double carbon sp${}^{2}$ bonds forming its honeycomb lattice. The stiffness, toughness, and strength of graphene has been leveraged to create novel composites with graphene as the reinforcing agent [15]. Lee et al. used an atomic force microscopy (AFM) nanoindentation setup to experimentally measure the second-order elastic stiffness, E2D as 340 ± 50 N m^−1^. This corresponds to a Young’s Modulus of 1.0 ± 0.1 TPa assuming a monolayer thickness of 3.35 Å. Lee et al. also measured an ultimate tensile strength of pristine graphene of up to 130 GPa [1] which marked graphene as the strongest material ever measured. Due to high Young’s Modulus and low mass of atomically thin graphene, graphene resonators can achieve mechanical resonant frequencies of 1.17 GHz and frequency tunability by high stress-strain manipulation up to 10${}^{3}$ MPa for membranes and 400 MPa for ribbons [16,17].

### 3.2. Electronic Properties

The initial excitement and popularity of graphene as an exceptional 2D material was a result of its unparalleled electronic properties [10]. The high electron mobility and electrical conductivity of monolayer graphene are a result of the delocalized pi bonds above the primary 2D plane which are formed because of the sp${}^{2}$ hybridization of the layered structure of graphene [13]. Graphene is considered a unique zero bandgap semiconductor as its valence band and conduction band coincide at the Dirac points. At room temperature, the electron mobility of sub-5 layer graphene has been measured to be above 15,000 cm${}^{2}$ V${}^{-1}$ s${}^{-1}$ [10], while some monolayer studies have estimated it to be above 200,000 cm${}^{2}$ V${}^{-1}$ s${}^{-1}$ near absolute zero temperatures [18]. The electronic properties of graphene have also been leveraged to create highly conductive nanocomposites (polycarbonate, polyester, polyurethane, polyaniline etc.) [19].

### 3.3. Thermal Properties

The ability to extract heat out of micro/nano-devices critical in many applications, including in MEMS/NEMS. The thermal conductivity of monolayer defect-free suspended graphene as measured by Balandin et al. was in the range of 4840–5340 W/mK [20]. Seol et al. [21] further explored the effect of contact with a Si/SiO${}_{2}$ substrate on the thermal conductivity of exfoliated monolayer graphene. They found that the thermal conductivity of this configuration of graphene was 600 W/mK which is significantly lower than that of suspended graphene and carbon nanotubes (3000 W/mK), but still exceeding that of the bulk materials typically used in microelectronics.

For applications in integrated MEMS/NEMS and electronic devices it is important to understand the coefficient of thermal expansion (CTE) of graphene. The theoretical first principles estimates show that graphene has a negative CTE at temperatures up to 2500 K [22]. Bao et al. used an electron microscopy approach to measure a transition from negative to positive CTE after 350 K [23]. Yoon et al. used Raman Spectroscopy to extract a negative CTE in the range of 200–400 K [24]. These unique CTE properties may enable future designs of thermomechanical devices that utilize graphene as the primary structure.

### 3.4. Electromechanical Properties

The centrosymmetric geometry of graphene means that the piezoelectric effect is not present in pristine films [25]. This inversion symmetry must be broken to generate piezoelectricity. Ong et al. showed that this can be done by doping on side of the graphene. Wang et al. [26] and Rodrigues et al. [27] strain engineered graphene in the in-plane and out-of-plane directions to observe the piezoelectric effect. The piezoelectric activity can be attributed to its ultrathin structure and chemical interaction with the substrate underneath, and was found to be twice as high as that of ceramics in the zirconate titanate family [27].

The sensitivity of piezoresistive sensors increases with larger sensing area and lower membrane thickness, which can lead to orders of magnitude higher sensitivities for graphene sensors [28]. Graphene-based strain sensors have the capability to outperform Si-based and other carbon material-based sensors due to higher force sensitivity and gage factors (GF) [11]. Hosseinzadegan et al. [29] reported the GF of graphene on Si_*x*_N_*y*_ membrane to be as high as 18,000, while those of conventional Si devices ranges from 30–40.

## 4. Graphene in MEMS/NEMS

Microelectromechanical Systems (MEMS) have been extensively researched since the 1960s, and the commercial viability of MEMS devices has been enabled by the development of standard silicon surface micromachining techniques [30,31,32,33,34]. With multifunctional capabilities like sensing, actuation and communication, MEMS devices like pressure sensors [35,36], accelerometers [37,38,39,40], and micromirror arrays [41,42,43,44] are ubiquitous in aerospace, biomedical and consumer electronics industries. The maturity and cost-effectiveness of these simple yet powerful devices are primarily driven by the integration of mechanical and electrical elements fabricated in a top-down manner using Si-based microfabrication techniques. However, with further demands for miniaturization, fabrication of devices in the submicron length scales integrated with microelectronics falls in the regime of nanoelectromechanical Systems or NEMS [45,46,47]. Unlike most existing MEMS manufacturing approaches, NEMS require the integration of 2/2.5D nanostructures with microelectronics which can be classified as a bottom-up approach [46,48]. Due to contradictory fabrication approaches between nanostructures and conventional MEMS/NEMS devices, integration of MEMS/NEMS devices with 2D structured graphene is challenging.

### 4.1. MEMS Switches

A useful application of MEMS/NEMS devices are radio frequency (RF) switches used in mm-wave communication systems [49,50,51,52]. Similar switches are used in non-volatile memory (NVM) devices [53,54,55], infrared imaging systems [56,57,58], chemical/gas sensors which require high frequency switching [59,60].

Graphene field-effect transistors with high on-off ratios [61] are also used for mechanical switching devices. A dual polarity graphene NEMS switch can be utilized as electrostatic discharge (ESD) protection structure [62] as shown in Figure 1.

However, realizing reliable elecromechanical coupling with high bandwidth and low-loss resonant structures is a major challenge in designing MEMS/NEMS for these applications [63,64,65,66]. For higher switching frequencies, MEMS switches need to be thinner, stiffer, and lower mass [67]. Silicon-based switches have lower limits of thickness and internal stress which limit performance. This is where carbon-based materials like carbon nanotubes (CNTs), diamond, and graphene outperform Si-based MEMS/NEMS devices.

### 4.2. Mass Sensors

Graphene resonators have been used as mass sensors [68,69,70,71,72], charge sensors [73,74,75] and extremely sensitive force sensors [76]. The 2D structure of graphene resonators, high surface area-to-mass ratio, and high operating frequencies make graphene resonators reliable for high precision mass sensing [11]. While state-of-the-art NEMS sensors can detect up to a single molecule mass [77], graphene nanoribbon based resonators have the potential to detect the mass of single protons, in yoctogram resolution [78]. For mass sensing applications, the correlation between addition of an external mass and a shift in the resonator’s natural frequency is called mass responsivity, which is the negative ratio of the resonant frequency to the effective mass of the resonator. Therefore, for higher mass responsivity, either the resonant frequency must be increased or the effective mass of the resonator must be reduced [79,80,81,82]. A higher mass responsivity also corresponds to a lower detectable mass (or mass resolution). The theoretical and experimental range of the responsivity of graphene mass sensors has been reported between 10${}^{15}$–10${}^{30}$ Hz/g [71,83,84], while Si-based sensors typically exhibit a responsivity of 10${}^{10}$–10${}^{18}$ Hz/g [80,85]. Additionally, the ability to control and tune the frequency of graphene sensors at room temperature significantly improves its mass sensing capabilities, as found by Singh et al. [86]. For force sensing applications, the force sensitivity is inversely proportional to the square root of Q factor and resonant frequency. In other words, graphene sensors exhibit the best force and charge sensing capabilities at higher Q factors which often require mK range temperatures [11,86]. Bunch et al. calculated the force sensitivity and charge sensitivity of a doubly clamped graphene resonator to be 1 fN/Hz and 80 × 10^−5^ e/Hz, respectively, for a resonant frequency of 35.8 MHz and Q factor of 100 at room temperature [68].

### 4.3. Pressure Sensors

The piezoresistive property of graphene NEMS can be leveraged for pressure sensors [87,88,89,90], strain gauge [91,92,93,94] and accelerometer [95,96,97,98] applications. The higher sensitivity of these sensors is due to the order of magnitude difference in the area and thickness of stacked or suspended graphene compared to their silicon counterparts. Smith et al. [99] developed a pressure sensor using suspended graphene and achieved a gage factor of 3.67. Zhu et al. used the piezoresistive property of multilayer graphene on a Si${}_{3}$N${}_{4}$ membrane to design a pressure sensor operating between 0–700 mbar [28]. Biomedical application of piezoresistive graphene has been utilized in catheters [100]. Resonant graphene drums can also be used as pressure sensors where shift in resonance frequency is measured by interferometry [101]. The pressure measured from this work can detect pressure changes from 8 to 1000 mbar with 4 Mhz frequency shift as shown in Figure 2. Electrostatically coupled graphene drums can measure from 25 to 1000 mbar [16] as shown in Figure 3. Graphene intracranial pressure sensor which is designed for the use in biomedical applications has been simulated to measure from −13 to 250 mbar with 1 mbar resolution [102].

Recently, graphene field-effect transistors have been implemented as pressure sensors. These pressure sensors are typically designed to cover a wide range of pressures for tactile sensing and it has been reported to measure pressure changes from 2.5 to 30,000 mbar [103].

For sensing strain, Zhao et al. [104] fabricated sensors with graphene islands on a mica substrate which led to significant changes in charge tunneling under strain, thereby leading to high gage factors (~300) and higher sensitivity. Other hybrid approaches using graphene composites have also been used to fabricate high sensitivity strain sensors [105].

### 4.4. Other Applications

Apart from resonators and piezoresistive sensors, suspended graphene has also been used to propose nanoscale generators [26,107], energy harvesters [108] and actuators [27]. Furthermore, graphene membranes can be engineered to have a high degree of porosity to allow them to act as filtration medium for desalinating seawater [109] and separating different gases [110,111]. The membrane porosity can be modified and controlled using additional steps like ion beam and electron beam bombardment [112,113,114] and chemical etching [115]. The combination of porosity and electrical conductivity of graphene membranes can be used to translocate and sequence DNA strands with much higher resolution, speeds, and reduced costs as shown in the works of Merchant et al. [116] and Garaj et al. [117].

## 5. Scaling Graphene Production

Before graphene can be used in any commercial MEMS/NEMS devices described in the previous section, large-scale graphene manufacturing processes must be implemented. The majority of existing literature on characterizing graphene properties and MEMS/NEMS focuses on small, lab scale samples which can not be produced with enough throughput for industry. There are many promising approaches to growing and transferring graphene on a large-scale but first the major limitations of any such processes must be understood.

### 5.1. Manufacturing

Graphene was first isolated through mechanical exfoliation in 2004 [118]. This process begins with bulk graphite. First, adhesive tape is used to mechanically cleave the topmost layers from the crystal. Then the tape is brought into contact with a silicon wafer to transfer the flakes that have been cleaved from the bulk crystal. Upon examination under an optical microscope, monolayer flakes can be observed and then patterned into experimental devices. These monolayer flakes can be several μm${}^{2}$ and very high quality [14]. However, this process is not scalable for practical devices.

The next major advancement in graphene synthesis was growth on catalytic metal foils. Li et al. pioneered this work with monolayer graphene on copper foils [119]. Copper is a desirable growth substrate because its carbon solubility is very low, so the growth process is surface limited, resulting in continuous single layer films. Researchers then began to try other catalysts such as platinum [120], Nickel [121,122], and even insulators such as sapphire [123]. By changing the growth catalyst and other process parameters, the thickness and crystallinity of the films can be tuned. Chemical Vapor Deposition (CVD) is a standard, scalable process in the semiconductor industry and can be well controlled, so it is highly desirable over mechanical exfoliation.

While CVD has potential for being scalable, there are still some downsides to the resultant films compared to exfoliated films, namely grain boundaries and wrinkles. Exfoliated graphene benefits from being inherently single crystal however CVD graphene has grain boundaries that appear when the independent domains begin to stitch together [124]. Grain boundaries scatter the electrons travelling through the material and release energy in the form of heat as well as provide defects for fracture to initiate. Previous studies suggest that electrical conductance improves with larger grains [125] and better stitching between domains in CVD graphene [126]. There are also some differences in mechanical properties which will be discussed below.

Graphene holds promise as a mechanical material in Microelectromechanical systems (MEMS) due to its extremely high elastic modulus and strength. Elastic modulus of exfoliated and CVD graphene is similar in many cases, which is around 1 TPa or 340 N/m ($E_{2D}=E\times t$). Figure 4 shows results comparing CVD and exfoliated graphene. The 2D elastic modulus is the same for pristine, large, and small grain graphene. It is worth noting that spread in the data presented here is large. On average, devices have similar properties but values can vary up to ±35%. One difference between monocyrstalline, exfoliated graphene and polycrystalline CVD graphene is the differences in fracture strength [127,128]. Comparisons between CVD and exfoliated graphene reported in literature are cited in Table 1. It can be seen that CVD graphene has slightly degraded mechanical properties as compared to exfoliated graphene, but the young’s modulus and fracture strength are still exceptionally high compared to conventional materials. The benefits of scale using CVD outweigh the costs of costs of slightly degraded mechanical properties, as long as the altered properties are known and repeatable.

Other potential benefits of CVD over mechanical exfoliation is the ability to precisely control layer thickness. In certain applications, bilayer [134,135], and even multilayer graphene may be desired [136]. Multilayer graphene has the benefit of being more durable during processing and can be manufactured in much the same way as single layer graphene, but with slight modifications to the process [137].

Another important mechanical property for MEMS materials is fracture toughness. Fracture toughness is the measure of a material’s resistance to crack propagation. Graphene performs poorly in this respect due to its 2D nature. Cracks formed at defects in the single crystal films can propagate without anything to arrest their growth. However, the presence of grain boundaries in CVD graphene may actually inhibit crack propagation and improve the fracture toughness of the films. Applications which require high fracture toughness would be devices which receive high amplitude cyclic loading, such as a resonator or ultrasonic transducers [138].

Since graphene is a brittle material, the ultimate tensile strength will ultimately depend on fracture toughness. Therefore, this metric must be understood as production scales to create repeatable films. Molecular simulation and limited experimental work has been done to determine the mode I fracture toughness of pristine, single crystal graphene as well as polycrystalline CVD graphene. Simulated toughness of pristine single crystal graphene ranges from 0.13–33.18 J/m${}^{2}$ or 3.3–4.7 MPam${}^{1/2}$ while the toughness of polycrystalline graphene ranges from 8–20 J/m${}^{2}$ [139]. It was accurately demonstrated that the fracture toughness of polycrystalline graphene is strongly characterized by strength in grain boundaries as cracks are most likely to propagate through the grain boundaries [140]. Zhang et al. were the first to measure CVD grown polycrystalline graphene and found a value of 15.9 J/m${}^{2}$ or 4.0 ± 0.6 MPam${}^{1/2}$.

One method researchers are exploring to improve toughness in polycrystalline graphene is increasing the number of layers. Wei et al. determined the fracture toughness of multilayer graphenes to be 12.0 ± 3.9 MPam${}^{1/2}$ [141]. They showed that the multilayer stacks have rotational mismatch with respect to each other, therefore a fracture in one layer is unlikely to align with the armchair or zig-zag direction which are the weakest directions. Figure 5 shows a comparison of fracture toughness for single layer and multilayer graphene. For graphene, the mulitlayer material had over three times the fracture toughness of its monolayer counterpart. Boronitrene also showed a doubling in fracture toughness in multilayer form. This finding motivates the use of using multilayer graphene in highly demanding applications that require a tougher material.

### 5.2. Intrinsic Defects

When scaling graphene production using CVD on large scales, different types of defects at different length scales begin to affect the performance of the MEMS/NEMS devices in which they’re implemented. Intrinsic defects are present in the graphene films once they are grown on their target substrates. Depending on the chosen growth method, device size scale, and type of device, the importance of each type of defect will change. The major defect types that will be discussed are graphene grain boundaries, wrinkles, and growth substrate morphology.

#### 5.2.1. Graphene Grain Boundary Effects

Graphene grown using CVD is inherently polycrystalline and includes grain boundaries. These boundaries are where graphene domains stitch together and may form incomplete bonds at different orientations. Typically adjacent grains have some angular misalignment and end up forming pentagons and heptagons instead of the traditional hexagonal bonding patterns [142,143]. According to computational simulations, the angle of two angular misalignment planes can enhance or deteriorate overall strength of graphene depending on the angle of the misalignment [143].

Suk et al. investigated the effect of grain size on fracture strength of single layer graphene [128]. Figure 6 reports the failure strength of single layer graphene when stressed using an AFM as a nanoindenter. To determine the size of grains, the group used carbon isotope labeling. ${}^{13}$CH${}_{4}$ and ${}^{12}$CH${}_{4}$ were used at different points in the growth process in order to label individual grains. The characteristic Raman peaks of the ${}^{13}$C and ${}^{12}$C are distinguishable and can be used to determine grain size quickly and without atomic resolution imaging. As expected, they found that a higher density of grain boundaries reduces the failure strength of the suspended films. Equation (Equation 1) is the model the authors used to calculate maximum stress, and it requires elastic modulus as an input. However, the authors in this investigation did not measure modulus independently, so they were forced to assume a value. Figure 6a plots the failure strength depending on two different moduli previously reported. Suk et al. denote small grains as <5 μm and medium grains as 10–20 μm in lateral dimensions.
(1)$$\begin{array}{c} F=\frac{1}{h}\sqrt{\frac{PEh}{4\pi R}} \end{array}$$

Later on, the same authors also measured modulus of similar 8–9 μm diameter membranes using bulge tests [132] and found that the elastic modulus of the graphene they measured depended on the grain size. Figure 7 shows the distribution of modulus with respect to GB density. Single crystal and medium-grain graphene was shown to have similar modulus to pristine graphene, but small grains had a modulus of less than half of what was expected. This finding can be explained by the existence of grain boundaries themselves. It has been shown that GBs induce out of plane corrugations and these wrinkles effectively reduce the stiffness of the suspended membrane [130]. Additionally, this softening depends on the graphene grain size. The larger the grains, the smaller the corrugations [144]. Further discussion on wrinkles is presented in Section 5.2.2.

Since material properties are dependent on growth process conditions, it is important to set control limits to ensure uniformity. CVD grown graphene will need to have predictable, repeatable mechanical properties in order to be scaled for industral applications. Work must be done to reduce the effect of grain size, compensate for the shortcoming, or design processes which can produce single crystal materials to sidestep the problem entirely.

Lee et al. performed a similar comparison of strength and modulus in CVD graphene [127]. They found that modulus is virtually the same and the fracture loads are slightly degraded by a larger grain boundary density. Figure 8 shows these results. Once again, these differences a slight and may be missed if the sample size is too small. Lee et al. denotes small grain as grain size from 1–5 μm and large grains as 50–200 μm (compared to small grains as <5 μm and medium grains as 10–20 μm by Suk. et al.). This study didn’t find a significant degradation in modulus like the previous studies mentioned. This could be due to much smaller diameter holes used (1 and 1.5 μm vs. 8–9 μm). With smaller diameters, there are fewer grain boundaries. Additionally, molecular dynamics simulations have shown that larger grains have less corrugations induced into the membrane at their boundaries [144]. An alternative explanation could be that the processing methods degrade properties by introducing defects in the copper etch and polymer scaffold removal step. This possibility is further discussed in Section 5.2.3.

Up until now, grain boundaries discussed have been assumed to be completely stitched together. However, this may not always be the case since grains are sometimes only overlapping. Overlapping grains may appear as a wrinkle, but when observed under TEM, the grain orientation of either side of the boundary are different [127], as seen in Figure 9. These overlapping boundaries are weak since only van der waals forces are holding the boundary together. Figure 9E shows the force vs load curve of overlapping grain boundaries. The blue curve is the expected behavior, but since the boundary was only bonded with van der waals forces, the grains simply slipped past each other at low loads. Lin et al. found in nanoindentation tests that suspended CVD monolayer graphene could be sorted into two categories: weak and strong [133]. Figure 10 shows the bimodal distribution of modulus and fracture force in single layer graphene. This finding could be indicative of the difference between weakly and strongly stitched grains. The authors reported that devices with weakly stitched grains failed at very low loads after the first or second loading cycle. One would not expect the Young’s modulus to depend on an inelastic event like the failure of weakly stitched grains. One possible explanation is a lower “effective” modulus measurement due to the lower quality data from the early failures of the weakly stitched devices. Elastic modulus is extracted from the non-linear stiffness at high deformations, and if the membrane isn’t deformed enough, not enough data is collected in that regime and the curve fit will be poor, leading to lower recorded modulus values.

An interesting finding is that overlapping grains enhance electrical conductivity compared to stitched grain boundaries [126,145,146]. This will benefit electrical applications, but is not very useful for mechanical applications since these boundaries are very weak. This also raises questions about the efficacy of using electrical resistivity as a measure for graphene quality. Generally, as graphene quality increases, the stitching between grains improves and resistivity decreases [147]. However, since Tsen et al. showed that overlapped grains have higher conductance, this result might not always hold true.

#### 5.2.2. Wrinkles and Ripples

Graphene exhibits out-of-plane surface corrugations when examined microscopically. These corrugations are primarily classified as ripples and wrinkles based on their topology, dimensions and aspect ratios. Ripples are isotropic peaks and troughs with critical height dimension nominally below 10 nm. On the other hand, wrinkles are high aspect ratio corrugations with width and height below 10 nm, but with lengths exceeding many μm. These corrugations impact the electronic, mechanical, optical, and chemical properties of graphene [148], and are inevitable in a high throughput production environment. Therefore, a better understanding of the different mechanisms causing ripples and wrinkles must be developed to control their formation and leverage the modified physical properties when implemented into MEMS/NEMS devices.

Chae et al. [149] investigated the origins of different wrinkles in graphene transferred onto Si/SiO${}_{2}$ substrates. They began with depositing thin Ni films onto Si/SiO${}_{2}$ wafers and grew graphene on the thin metal films using CVD. After growth, the graphene films were transferred onto bare Si/SiO${}_{2}$ wafers for analysis. They noticed 2 main types of wrinkles, which are shown in Figure 11. The first type of wrinkle is a closed wrinkle that forms around the grains in the catalytic metal substrate. The bright vertical line in the left center Figure 11 is a good example. This type of wrinkle is the most prominent and depends on graphene growth substrate morphology. Liu et al. showed that there is a correlation between substrate grain size and the density of this type of wrinkle [150]. In other words, the larger the metal grains, the fewer graphene wrinkles attributed to them.

The second type of wrinkle is a purely thermal stress-induced wrinkle. The thermal coefficient mismatch of the graphene and metal films create a significant internal compressive stress, which is then relieved by local buckling of the films. These wrinkles are generally short and formed in straight lines within the confines of the graphene “islands” created by catalytic metal grain structure. These types of wrinkles are circled in Figure 11. N’Diaye et al. successfully imaged graphene in-situ using low energy electron microscopy (LEEM) during the growth process to show the onset of these wrinkles [151]. They grew graphene on Ir(111) substrates at 1110 °K, and Figure 12 shows the formation of wrinkles. The evolution from sub-figure I-II occurred in one second at 560 °K. The insets show the formation of thermal stress induced wrinkles on the left side of images I–II. Raman studies have shown that once these wrinkles have formed, and can be local non-uniform strain distributions at scales on the order of 100 s of nm [152]. However, strains may become uniform over µm sizes.

Figure 13 shows the mechanism of wrinkle formation in multi-layer graphene along defects (step edges and grain boundaries) in the metal catalyst. Step edges in the metal substrates favor graphene growth [153], so graphene commonly nucleates there. When grains overlap at these locations, wrinkles can form, which is shown in Figure 13a. It should be noted that while this reference specifically discusses few-layer graphene films grown on nickel substrates, these wrinkle formation mechanisms apply to single layer graphene grown on catalysts such as copper because the thermal expansion coefficients are similar in both metals. While cooling, the significant differences between the coefficient of thermal expansion for graphene (−8.0 × $10^{-6}$ K${}^{-1}$) [24] and the catalytic growth material, typically Cu (20 × $10^{-6}$ K${}^{-1}$) [154] or Ni (13.3 × $10^{-6}$ K${}^{-1}$ [155]) causes wrinkles to form at defects and boundaries in the metal catalyst. This is because after cooling from a typical growth temperature of 1050 °C, the metal contracts while the graphene slightly expands. This causes a compressive strain of 2–3% in graphene [150], which buckles the graphene and creates the wrinkles seen in Figure 13b.

Ripples, on the other hand, are smaller in scale and can be harder to measure. Surface adhesion forces can obscure ripples on substrates, therefore ripples are best observed while suspended. Figure 14b is a representation of these ripples and the scale of this roughness is matched qualitatively [156]. An example of the physical ripples present experimentally is shown in Figure 14d. These small scale ripples are a result of multiple factors. First, graphene is grown on an inherently rough metal substrate which transfers its roughness to the graphene [157]. This roughness can lead to additional rippling once edge constraints are imposed onto the membrane. Additionally, nonuniform adhesion between the membrane and the edge of the holes can cause anisotropic pretension which could transfer shear strain into the membrane to cause ripples [158]. Finally, the presence of defects and grain boundaries in the graphene itself is predicted to induce surface topography [142,144,159], shown in Figure 14c. Many atomic simulations show this effect, but it is likely that this grain boundary effect is the smallest of the three just mentioned.

While they can occur for a number of reasons, all wrinkles and ripples effectively soften graphene’s elastic modulus since energy is being used to flatten out the out-of-plane features instead of stretching the membrane [130]. Preliminary work performed by Lin et al. has shown that when these wrinkles at the grain boundaries are smoothed out, the reduction in modulus can be reduced or completely eliminated in some cases [133,160]. Figure 15 shows that after repeated load cycling with an AFM nanoindenter, the membrane has a visible increase in surface area induced while increasing material stiffness. The authors performed image analysis around the boundary of the membranes and found no evidence of slipping, and previous studies have proved the high adhesion energies of graphene on SiO${}_{2}$ surfaces [161]. Therefore, this effect can be attributed to smoothing of wrinkles. Along with this increase in slack, the local surface roughness is reduced slightly from 1.12 to 0.95 nm, further suggesting a reduction in the wrinkles. The increase in surface area shown on the left of Figure 15, should not affect the calculated membrane stiffness because the warping will be suppressed by a small initial elastic stretch in the first stage of the nanoindentation [143].

Finally, graphene grain boundaries aren’t necessarily shared with the metal catalyst grain boundaries. Single graphene grains have been shown to be continuous along steps, positive and negative edges, and positive and negative vertices [157]. This is promising and means that pristine graphene isn’t limited by the underlying catalyst structure. However, as the different graphene grains stitch together, they recreate the 3D geometry of the substrate in the graphene film itself. Additionally, Chae et al. also showed that smaller, thermal stress induced wrinkles or step-terrace like edges can cross over the metal catalyst grain boundaries [149]. This indicates that the grain boundaries cannot play a role in nucleation, potentially due to its lower carbon solubility. This is good as allows graphene grains to grow larger independent of the underlying catalyst grain structure, creating stiffer films, as described in Figure 7c. This could be due to a reduction in wrinkle density.

#### 5.2.3. Voids and Other Defects

Researchers have found that standard process steps can significantly reduce the fracture strength and toughness of graphene. Specifically, etching copper in ferric chloride (FeCl${}_{3}$) and annealing the graphene in air to remove the supporting polymethyl methacrylate (PMMA) layer have been shown significantly weaken the graphene [127]. This is because metal particles introduced while etching the metal substrates used for growing graphene have been shown to etch graphene which introduces defects into the graphene structure which can then be chemically activated during the anneal step. [162,163,164]. If ferric chloride is replaced with ammonium persulfate ((NH${}_{4}$)${}_{2}$S${}_{2}$O${}_{8}$ or APS) and the anneal step is removed by using a Polydimethylsiloxane (PDMS) dry stamp transfer method, fracture loads very closely match the reported loads for pristine graphene.

### 5.3. Importance of Repeatability

After scalability is addressed, the next issue is assuring the quality of the large area films produced. Repeatability of the manufacturing process will depend on the methods selected, but there are many common defects and potential weak points throughout all processes. Fundamentally, repeatability means that all areas on a given large area substrate should produce devices within an acceptable tolerance for the given performance metric. Different performance metrics are affected by different defects. For example, some of graphene’s outstanding electrical properties are only present when graphene is only a single layer thick since multilayer areas scatter electrons. However, it is often extrinsic defects can are introduced to graphene during the patterning and transfer steps that are required to manufacture devices that cause the largest issues with repeatability.

#### Methods of Graphene Transfer

A majority of applications of graphene require transfer from the growth substrate onto the target substrate. There are numerous transfer methods which can be categorized by the usage of supported polymer layer which must be dealt with using solvent to remove polymer layer.

The examples of the methods are polymer-supported transfer including wet etch transfer [165,166], electrochemical delamination transfer [167,168], and direct delamination [169]. Polymer-free transfer utilizes alternative way to delaminate graphene from the substrate [170,171,172].

Wet etch transfer typically refers to transfer by spin-coating a supportive polymer layer (PMMA, PDMS, etc) then etching away the metal substrate. Once supportive polymer layer with graphene is cleaned and transferred onto Si substrate, the sample is dried and annealed to improve adhesion between graphene and the substrate. Afterwards, supportive polymer layer is removed using solvent. This process is first reported for graphene grown using CVD [173] and overall process diagram illustrated in Figure 16.

In general, polymer-supported wet etch transfer is widely used for numerous applications. It is a convenient way to process graphene and effectively smooths and flattens the graphene onto the substrate. However, removal of polymer after transfer can be challenging as any residual polymers left on the surface tends to deteriorate properties of graphene. According to Wang et al. [174], graphene devices processed with different transfer techniques such as polymer-supported wet transfer, etch-based transfer with residual doping, and direct delamination show different sheet resistances. The sheet resistance for graphene produced using various transfer techniques is shown in Figure 17c with a comparison to an ITO film on PET which is commonly used in transparent electrodes. The result tells us that as more polymer layers are used during the transfer process, higher sheet resistance will be observed on graphene electronics. This is due to the residual polymer having p-doping effect on graphene [175]. Electron mobility is also known to be dependent on the transfer process [176]. Polymer-free transferred graphene exhibits 30–50% higher electron mobility on average compared to graphene from polymer-assisted transfer. Polymer-free transferred graphene showed electron mobility as high as 63,000 cm${}^{2}$ V${}^{-1}$ s${}^{-1}$ [171].

On the other hand, direct delamination, or dry-transfer, can be achieved by directly peeling graphene off of the growth substrate and relocating it to the target substrate. The transfer utilizes rate dependence of adhesion forces to overcome traction energy between graphene and the metal substrate [169]. Additionally, functionalization of surfaces between graphene and the target substrate can enhance the adhesion between two surfaces. This process eliminates potential deterioration of mechanical, electrical, and chemical properties of graphene while using polymer-assisted transfer [174]. Therefore, direct delamination provides consistent, high quality, and fast transfer of graphene compared to polymer-supported transfer which may be suitable for roll-to-roll nanomanufacturing [177].

As shown in Figure 17b, transfer without the usage of PMMA exhibits lower sheet resistance. There have been efforts to transfer graphene without a supportive layer to minimize deterioration of electrical properties. One of the efforts is dry transfer which is a method which utilizes differences in adhesion energy of adhesives or polymer stamps without the use of solvents. A fast pick up method using PDMS stamps to remove a graphene monolayer out of bilayer graphene is an example of one of these dry transfer methods [178]. Stamp transfer utilizes adhesion energy difference between PDMS-graphene and graphene-graphene interfaces to pick up one layer of graphene from a multilayer graphene structure. Single crystalline graphene can be easily worked around with the stamp transfer; however, for polycrystalline graphene, the technique tends to yield failures on graphene grain boundaries as adhesion energy between graphene-substrate can be higher than between grain boundaries. Thus, these techniques are most effective on bilayer graphene as graphene-graphene interfaces have much lower adhesion energy compared to graphene-substrate interfaces.

Electrochemical delamination uses ion intercalation between polymer-graphene and growth substrate (metallic foil or thin-film) to separate graphene from the substrate [168]. This method does not require any etching of the metal substrate, hence metal layer can be reused for another graphene growth. However, the manual handling of the PMMA/graphene composite layers as well as hydrogen bubbles that form between the growth substrate and the graphene layer may potentially damage the graphene [179].

Polymer-free transfer eliminates the process of coating polymer on graphene and directly etches away the metal substrate. The problem of obtaining graphene without any kind of supportive layer has been resolved by utilizing hexane which traps graphene between hexane and metal etchant during release process and between hexane and DI water during the transfer process [170]. Other methods use a graphite holder to hold graphene as it is processed in a metal etchant solution [171]. Similar to this transfer, convex liquid is used to transfer graphene while it is floating on top the liquid [172]. These polymer-free transfer yields high quality graphene with state-of-the-art chemical, mechanical, and electrical properties. However, while all these polymer-free transfer process are suitable in laboratory scale research, scaling these processes will be difficult and sometimes not infeasible.

The main problem with having transfer included in the MEMS/NEMS graphene device fabrication is that yield is difficult to control. MEMS/NEMS graphene devices often have trenches or gaps between electrodes in order to enable the graphene to be suspend on the device but this requires alignment of the patterned graphene to the MEMS substrate. One of the methods to achieve this is pre-patterening the graphene structures so that they can be align-transferred to different substrates [180] as shown in Figure 18. However, this type of aligned transfer often introduces defects into the graphene such as localized strain or wrinkles. To avoid these transfer induced defects, direct-growth on transition thin-film metal has been proposed where growing graphene can be done selectively on pre-patterened transition metal thin-film and etching away the metal partially to obtain suspended graphene structures [181] as shown in Figure 19a. This concept can be applied similarly to the MEMS/NEMS graphene devices which requires suspended structure of graphene [8].

## 6. Proposed Solutions

Researchers are looking into ways to approach the problem of successfully manufacturing high-quality graphene at a large scale. Currently, the biggest challenges for mass produced graphene integrated MEMS/NEMS devices is having high quality graphene in large production scale with high throughput. Therefore, the aspects we considered important for proposed solutions are the ability to grow graphene over large areas, shorter growth times, maintaining the highest graphene quality, and improving the yield of the fabrication process. In order to summarize how current graphene growth research is being performed, we have plotted data from numerous references of growth time of graphene compared to electron mobility and grain size of graphene. As shown in Figure 20a, as growth time increases, mobility of graphene increases which shows us that the electrical quality of graphene is proportional to the growth time in most cases. Figure 20b shows growth time of graphene compared to average grain sizes of graphene. Grain size does not directly improve mechanical properties of polycrystalline graphene; however, it does reduce the prevalence of wrinkling in the graphene films, which improves device repeatability. Additionally, it has been reported that single crystalline graphene exhibits the highest mechanical strength of graphene since it is relatively defect free [182]. MEMS/NEMS device application using graphene must consider both of these electrical and mechanical properties as crucial. Therefore, our ultimate goal for MEMS/NEMS device applications using graphene requires shorter growth times while maintaining high mobility and large grain size which are highlighted in Figure 20a,b.

In this section, several active research areas will be introduced and discussed for further enabling of high-quality graphene for mass production. Mass production schemes discussed in this section includes graphene transfer process using roll-to-roll (R2R) processing, graphene growth and MEMS device fabrication on transition metal thin films for transfer-free process, controlled graphene growth on single-crystalline films, strain engineering on the substrate, and other approaches.

### 6.1. Roll-to-Roll (R2R) Processing

One of the most convenient and effective ways to reduce the cost of graphene production is to implement conventional manufacturing methods. Roll-to-Roll (R2R) processing is a well-known, high throughput process of making electronics on a roll of metal foil or flexible polymer. A variety of processes are available for coating, patterning, deposition, and etching in R2R manufacturing which can be performed continuously on a roll for miles at a time [4]. For larger area graphene production, R2R manufacturing methods using Cu foil has been implemented [188], as shown in Figure 21. Once graphene is grown and subsequently cooled down, graphene is coated with polymer that can assist with transfer process. There are two ways of working on polymer-assisted graphene transfer, with each having its strengths and weaknesses.

The first method is to etch the catalytic metal foil and roll the graphene on polymer into a separate roll for further processing. This process is conventionally known as wet etch transfer where only metal etchant touches the metal foil while graphene and the polymer are physically unstrained. The strength of this process is that graphene is exposed only to chemicals with which graphene is chemically inert. The weakness of the process is that the metal foil is not reusable as it is being etched away during the transfer process. In addition, polymer backing must be still be removed from the graphene for many applications [174].

The second method is known as dry peeling method where polymer-graphene layer is peeled off from the metal foil. Initially, graphene and the metal foil are bonded with Van der Wall’s force. Force of dry peeling must be higher than Van der Wall’s force between two surfaces in order to successfully delaminate the graphene from the metal foil. The force can be adjusted by the speed and the number of graphene layers peeled off from the metal foil [169,177].

The strength of R2R process is that the throughput can be high depending on the growth process. The rate at which graphene grows on Cu foil is determined by the precursor gas, the temperature of the CVD furnace, and surface reaction mechanism between the gas and the substrate for graphene growth. The speed of the graphene production through R2R production is superior to any other manufacturing technique currently in use for graphene transfer. Also, graphene characterization for quality inspection through R2R process is much easier via high-speed tip-based metrology inspection. This technique has been developed to inspect nanostructures over large surface areas; therefore, it has a potential to be utilized to observe graphene quality after the production [189]. In addition, transparent aspect of graphene allows colorimetry inspection of graphene which can further speed up the characterization time of graphene after R2R growth and the transfer [190,191].

However, in order to further improve cost-effectiveness, the yield of graphene devices, and the device performance, it is necessary to advance graphene production so that it is compatible within current semiconductor fabrication.

### 6.2. Transfer-Free, Graphene Growth on Thin-Films

The R2R transfer process provides superior speed of graphene growth over a large scale, but the process itself requires an additional step of transfer to the target surface. Adding another step to the process can be difficult for the application where the graphene has to be a suspend nanostructure. Graphene integrated MEMS/NEMS devices show higher mechanical and electrical performance when it is fully suspended compared to when it is operated on a substrate. However, graphene is exposed to capillary force during liquid-based transfer which often damages the graphene [170].

Transfer-free fabrication is one of the fabrication processes introduced to eliminate a step of using polymer backbone (typically poly (methyl methacrylate), or PMMA) to transfer graphene and instead grow graphene on the desired substrate [8,181]. Transfer process can physically damage graphene while being handled if the PMMA is not thick enough. In addition, PMMA-assisted transfer leaves residue on graphene which is responsible for deteriorating electrical properties of graphene. Thus eliminating this step improves the electrical performance of graphene devices, as shown in Figure 17 [174].

There have been efforts to eliminate the transfer process by growing graphene on metal thin-films directly above the devices they’re destined for. Patterning thin film allows graphene to grow on specified area and this can eliminate the necessity of patterning graphene for the device [8,17,89,179].

Using transition metal thin-films as a substrate for graphene growth is indeed a promising approach for application in MEMS/NEMS devices as depositing thin-film can be integrated into conventional MEMS/NEMS fabrication at the wafer scale. Depositing and patterning transition metal thin-films can also selectively grow graphene [192]. However, a majority of device layers used in MEMS/NEMS device applications are not compatible with the >1000 °C temperature requirement for graphene growth [193,194,195,196,197]. Therefore, there are not many examples of experiments which can fully integrate CVD growth with MEMS/NEMS devices yet. In addition, in order to cost effectively grow graphene on thin-metal-films, depositing the thinnest possible thin-film is considered ideal. However, the thermal stability of thin-films is reduced as the thickness is reduced. Graphene grown on thin films below 100 nm of Cu and Pt thin film thicknesses show that dewetting can occur at a temperature lower than its melting point [193,198,199]. Thermal instability of thin-films can be verified by Young’s Equation (Equation 2) where *R* is the average grain size, *t* is the thickness of film, and θ is the wetting angle of the metal to the substrate [200].
(2)$$\left(\frac{R}{t}\right)>\frac{3\;\sin^{2}\theta \;}{2-3\;\cos\;\theta +\cos^{3}\;\theta }$$

Eliminating transfer step sounds solid and appealing for ideal graphene fabrication process, but directly growing graphene on thin transition metal film is not trivial. There are several obstacles required to overcome to successfully grow graphene directly on a Si/SiO${}_{2}$ substrate. It is known that several transition metals do not have sufficient adhesion on Si/SiO${}_{2}$ substrates due to difference in electronic work function and require additional adhesion layers to maintain their integrity [179,201]. Cu thin film has been a good foundation of graphene growth on thin films and has shown device performance comparable to exfoliated graphene as long as crystalline structure of Cu thin film can be maintained in (111) [202]. However, the adhesion of thin film Cu is not sufficient on Si/SiO${}_{2}$ substrates which causes the thin film to dewet and deform even below the melting temperature of Cu [198]. Therefore, adhesion layers such as Ni, Ti, Ta, and Cr are required between the substrate and Cu layer. Additionally, the thickness of the Cu film can be increased to compensate for the instability.

Thermally stable materials such as platinum have been studied to improve the integrity of thin film during and after graphene growth. Pt is known to have a higher melting temperature and less lattice mismatch while maintaining self-limiting graphene growth due to platinum’s low carbon solubility. For this reason, thinner Pt thin film down to 100 nm can be implemented to successfully grow graphene which is difficult with Cu thin films [199].

Studies on selecting appropriate adhesion layers have also been reported where common adhesion layers of Ni, Ti, Ta, and Cr are used to support 300 nm Pt thin film to grow graphene. When the catalytic and adhesion layers are exposed to heat during graphene growth, those layers diffuse into each other and form an alloy. Therefore, alloying properties between the adhesion layers and the Pt thin film is important as imperfect alloying prevents uniform graphene growth on the metal thin film layer. The best alloying adhesion layer has been found to be Ni which uniformly alloys with Pt thin film at high temperatures [179,198]. The number of graphene layers can also change depending on the layer’s composition ratio between the adhesion layer and the transition metal layer. For Cu and Ni, as the ratio of Ni thickness over Cu thickness increased, more layers of graphene were grown on alloyed Cu-Ni thin film [185,198,203].

The substrate also plays an important role handling heat and affecting quality of graphene grown on top. For example, Pt thin film deposited on TiO${}_{2}$ on SiO${}_{2}$ significantly increases thermal stability, over Pt thin film on SiO${}_{2}$ substrate [193] as shown in Figure 22. Thermal surface dewetting was reduced maximum by 45 % when the Pt thin film was deposited on TiO${}_{2}$ as compared with direct deposition on SiO${}_{2}$.

Dewetting of a film can be prevented by either increasing film thickness or decreasing the wetting angle. As cost effectiveness of graphene growth process is inversely proportional to the film thickness, it is difficult to choose the appropriate film thickness for many applications. Therefore, we would like to introduce several ways to achieve low temperature growth in next section to provide alternative solutions to the obstacle.

### 6.3. Low Temperature Growth of Graphene

Low temperature growth of graphene is important to preserve the integrity of thin-film and devices that will be integrated with graphene such as MEMS/NEMS; however, it is not trivial to find appropriate ways to achieve low temperature growth. The first reason is that decomposition of certain carbon source gases require high temperature. Decomposition of common carbon source, methane or CH${}_{4}$, requires temperature up to 1050 °C on copper catalyst [204]. The second reason is that the growth temperature is directly related to the quality of graphene. Typically, annealing at higher temperatures for longer period of time increases grain size of metal which provides solid foundation for the graphene layer [198]. Also, grain boundaries of transition metal film act as a nucleation site which may lead to the growth of additional patches of graphene. As a result, it is important to minimize transition metal’s grain boundaries to successfully control the uniformity of graphene [183].

Several approaches to achieve low temperature graphene growth without sacrificing graphene quality have been attempted. Graphene growth utilizing plasma-enhanced/assisted CVD has shown to lower the growth temperature to 600 °C with low defect peak measured in Raman spectrum [195,205]. Without plasma, typical graphene growth on Ni found to be the lowest at around 800–900 °C [206]. Alternate carbon sources such as benzene, ethlyene, and acetylene have also been shown to enable lower temperature growth down to 300–400 °C depending on the substrate [207]. However, with all these approaches, the quality of the graphene is still generally below that of high temperature growth.

### 6.4. Production of High Quality Single Crystalline Graphene in Large Scale

The importance of the substrate for graphene growth has been known and emphasized as any non-uniform and defective substrate conditions tend to deteriorate graphene’s quality [202]. As discussed previously, graphene follows morphology of the metal catalyst where it is grown and maintains its integrity even after it is transferred to another substrate. When the growth substrate’s surface is defective, graphene may exhibit holes and damage throughout the film as reported previously [179,202]. Therefore, it is important to reduce surface roughness, impurities on the substrate, and lattice orientation mismatch between graphene and the substrate. Controlling surface roughness and crystalline structure is beneficial to control graphene growth mechanism as well as the nucleation density. If these parameters are not controlled, number of graphene layers grown on the substrate can vary as more carbon atoms accumulate on defects.

The effort to grow single crystalline graphene on foil has showed prominent progress for the last ten years focusing on increasing the grain sizes and reducing nucleation sites during growth. Grain boundaries in graphene are between two grains of graphene with different lattice orientations which produces incomplete carbon-carbon bonding [143]. Carbon atoms near the boundaries form SP${}_{3}$ hybridization which is a weaker bond than the SP${}_{2}$ bonding present in regular graphene. The degree of SP${}_{3}$ hybridization depends on angular alignment of the grains. The importance of growing single crystalline graphene arose as grain boundaries in polycrystalline graphene is known to affect electrical and mechanical properties of graphene [127,208,209,210,211,212]. As electron channel length is shorter, more electron scattering can be observed which can be critical for majority of electronic applications of graphene. Although polycrystalline graphene exhibits comparable mechanical strength and modulus to single crystalline graphene [127,143], the mechanical properties can be inconsistent as mechanical strength of grain boundaries dominate the overall properties of single crystalline graphene. For the consistency of MEMS/NEMS devices using graphene, forming uniform crystalline structure of graphene is one of the key parameters.

#### 6.4.1. Substrate Engineering for Single-Seed Growth for Single Crystal Graphene (SCG)

There are two approaches for growing single crystalline graphene over large areas. The first approach is single-seed growth. Single-seed approach starts graphene growth from single nucleation site and grows it until the whole transition metal substrate is covered. This approach has no graphene stitching from graphene from different grains. If multiple nucleation sites present, each grains will have its own crystal orientation, and these individually oriented grains have potential to form polycrystalline graphene when grain boundaries are formed. Downside of single-seed growth is that larger areas take longer to cover with graphene since the process time is governed by graphene growth rate from the nucleation site.

Eliminating majority of nucleation sites is challenging as the sites can be determined by the substrate’s roughness and other contamination. Reducing the surface roughness and the contamination can be done by pre-treatment process on the surface. For example, typical ammonium persulfate (APS) or acetic acid has been used to remove oxidation and other impurities on Cu foil surface [198]. Chemical mechanical polishing, annealing, and oxygen plasma have been shown to reduce the surface roughness and impurities on the surface. Substrate engineering plays an crucial role in how these nucleation sites are controlled. Graphene growth on single crystalline transition metal, such as Ru(111), Ir(111), Pt(111), Co(0001), Ni(111), Cu(111), and Cu/Ni(111) alloy has also shown promise in producing single crystal graphene [183]. These transition metal substrates require extra annealing steps to form single crystalline structures. As extra treatment and substrate engineering is necessary to obtain SCG using single-seed growth, this approach can be challenging and costly.

#### 6.4.2. Multi-Seed Growth for SCG

Multi-seed approach has been widely used in growing polycrystalline graphene. The strength of utilizing multi-seed growth is that graphene starts nucleation sites in multiple locations which results in covering the substrate at a much higher rate. The detailed growth mechanism is explained in Figure 23. This growth rate depends on the temperature and the precursor gas used during the growth. The challenge for multi-seed growth to obtain single crystal graphene across the substrate is maintaining the orientation of each graphene grains. One of the possible methods to achieve this goal is controlling growth conditions and crystallinity of the substrate. Experiments have been conducted where large grain domains on Cu(111) surface are promoted [213] to have aligned graphene grain domains across the substrate. Cu(111) and graphene have 4% lattice mismatch, and as temperature is highly elevated during graphene growth, the domains connect with each other creating seamless boundaries. It has been reported merging of grain boundaries is critical to electrical quality of graphene as electron scattering is observed near grain boundaries and wrinkles [5,184,214].

### 6.5. Other Approaches

As alternative solutions of using monolayer graphene grown from CVD for NEMS/MEMS devices, graphene oxide (GO) has been introduced to the field. GO can be produced in solution so it does not require a CVD process. Also, scalability of GO is feasible through modified Hummer method [215,216]. Preparing thick graphene oxide can increase the stability of the material, but it sacrifices the mechanical strength and electrical properties necessary for state-of-the-art MEMS/NEMS graphene devices [217].

The atomic structure of GO can be seen in Figure 24, which is not only composed with hexagonal carbon atoms but also with pentagonal and heptagonal carbon atoms [216]. The combination of incomplete carbon atom structure may yield lower mechanical strength compared to single crystalline graphene structure.

rGO drum resonators has been successfully utilized as nanomechanical devices in wafer scale as shown in Figure 25 [218]. rGO film is first coated using spin casting technique resulting in continuous film over the flat surface and then it is transferred to desired substrate. Using laser interferometry, Q-factor of a rGO drum resonator achiveved as high as 3000. One of the new approaches to utilizing GO for ease of fabrication has been shown with stamping transfer. Stamping transfer of reduced GO (rGO) has proven to show good transfer without using critical point dryer in liquid [219]. However, the minimum thickness of rGO is 5 nm and has lower mechanical strength. As a consequence, Q-factor and resonance frequency of suspended rGO resonators are lower than that of graphene. In addition, liquid phase reduced graphene oxide can be printed on the targeted location and polymerized by two-photon polymerization by direct laser writing [220]. This enables production of interesting features of coils in micron sizes. However, significant work still needs to be done for rGO to have similar electrical and mechanical properties to graphene and to make it a viable material for graphene-based MEMS and NEMS devices.

## 7. Conclusions and Future Perspectives

We have reviewed several approaches to overcome the current processing problems in manufacturing graphene MEMS/NEMS devices. MEMS/NEMS applications generally require suspended graphene structures, but current graphene growth and transfer has many limitations. Majority of graphene growth on foil must be transferred to another substrate. However, graphene may be contaminated or damaged during this transfer process. The polymers used in the process also generally leave residue during and after transfer which deteriorates mechanical and electrical properties of graphene. In addition, transferring graphene to another substrate often damages and wrinkles the graphene. These are extrinsic defects that are added on top of intrinsic defects of graphene from the growth process that must be accounted for to create repeatable devices. In order to minimize extrinsic defects, transfer processes need to be minimized and selective graphene growth on transition thin-metal films may become necessary. However, as graphene growth requires very high temperatures to produce high quality graphene, maintaining the integrity of thin-metal film is very important. Studies related to improving the integrity of thin-film in high temperatures have been conducted and proposed for feasibility of graphene growth on transition metal thin films that are less than 100 nm thick. Additional work has also been done to reduce the growth temperature to make graphene growth more compatible with conventional MEMS processing. Finally, engineering the substrate and growth process to produce single crystalline graphene at higher rates is a very important step for improved graphene-based MEMS/NEMS devices because the grain structure of the graphene has a significant effect of its performance in MEMS/NEMS devices. Therefore, while graphene shows great promise in MEMS/NEMS applications due to its electrical and mechanical properties, significant work still needs to be done on the manufacturing processes for integrating graphene into MEMS/NEMS devices before graphene-based MEMS/NEMS devicecs can become commercially viable.

## Figures and Tables

**Figure 1 micromachines-13-00027-f001:**
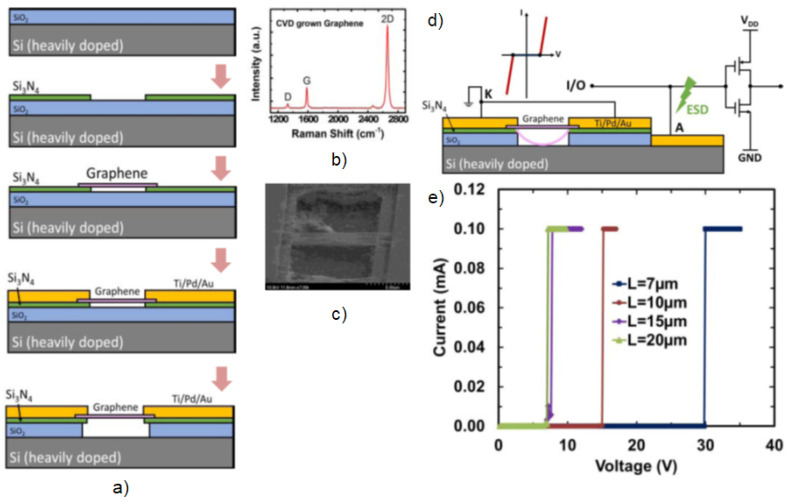
(**a**) Fabrication process of graphene NEMS switch. (**b**) Raman spectrum of suspended graphene structure. (**c**) SEM image of suspended graphene NEMS switch. (**d**) A cross-section of circuit diagram of graphene NEMS switch which protects ESD. (**e**) On-off ratio of graphene NEMS switch [62]. Reprinted (adapted) with permission from [62]. Copyright ©2016, IEEE.

**Figure 2 micromachines-13-00027-f002:**
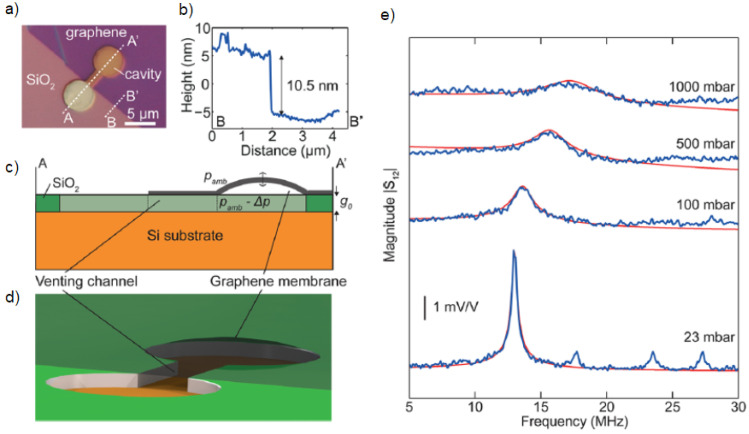
(**a**) Optical image of resonant multilayer graphene pressure sensor with cavities and venting channel. (**b**) Height profile of multilayer graphene from AFM measurement. (**c**) A schematic of squeeze-film graphene pressure sensor in cross-section view. (**d**) 3D view of graphene pressure sensor and cavities. (**e**) Frequency response of graphene resonator at different pressure levels [101]. Reprinted (adapted) with permission from [101]. Copyright ©2016, American Chemical Society.

**Figure 3 micromachines-13-00027-f003:**
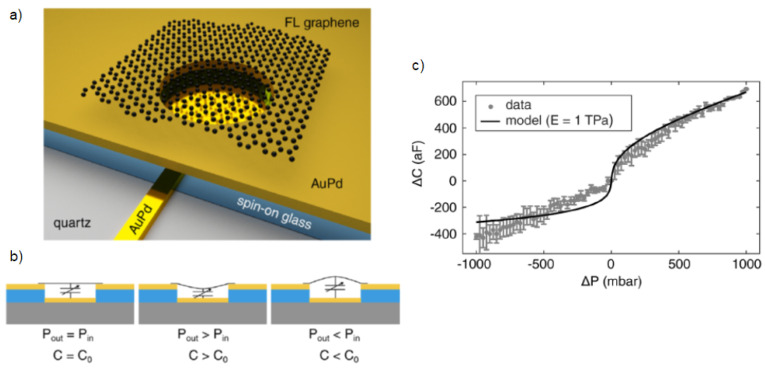
(**a**) A 3D schematic of capacitive graphene pressure sensor. (**b**) Images represent graphene at stationary, with attractive capacitive force, and with repulsive capacitive force applied to the electrode. (**c**) The graph shows capacitance changes according to the pressure change [106]. Reprinted (adapted) with permission from [106]. Copyright ©2017, American Chemical Society.

**Figure 4 micromachines-13-00027-f004:**
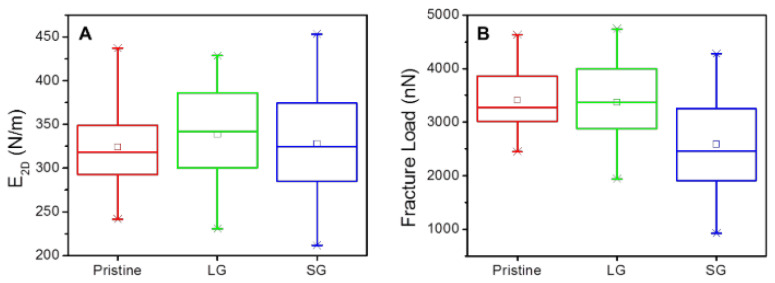
The box-plots of (**A**) elastic stiffness and (**B**) Fracture load for Pristine, Large grain (LG), and Small grain (SG) graphene [127]. Reprinted (adapted) with permission from [127]. Copyright ©2017, Emil Sandoz-Rosado et al.

**Figure 5 micromachines-13-00027-f005:**
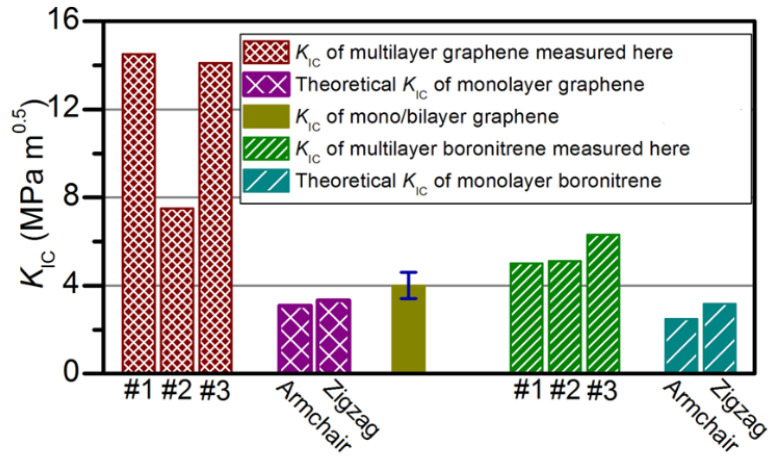
Comparison of fracture toughness between single and multi-layer graphene and Boron Nitrene [141]. Reprinted (adapted) with permission from [141]. Copyright ©2015, American Chemical Society.

**Figure 6 micromachines-13-00027-f006:**
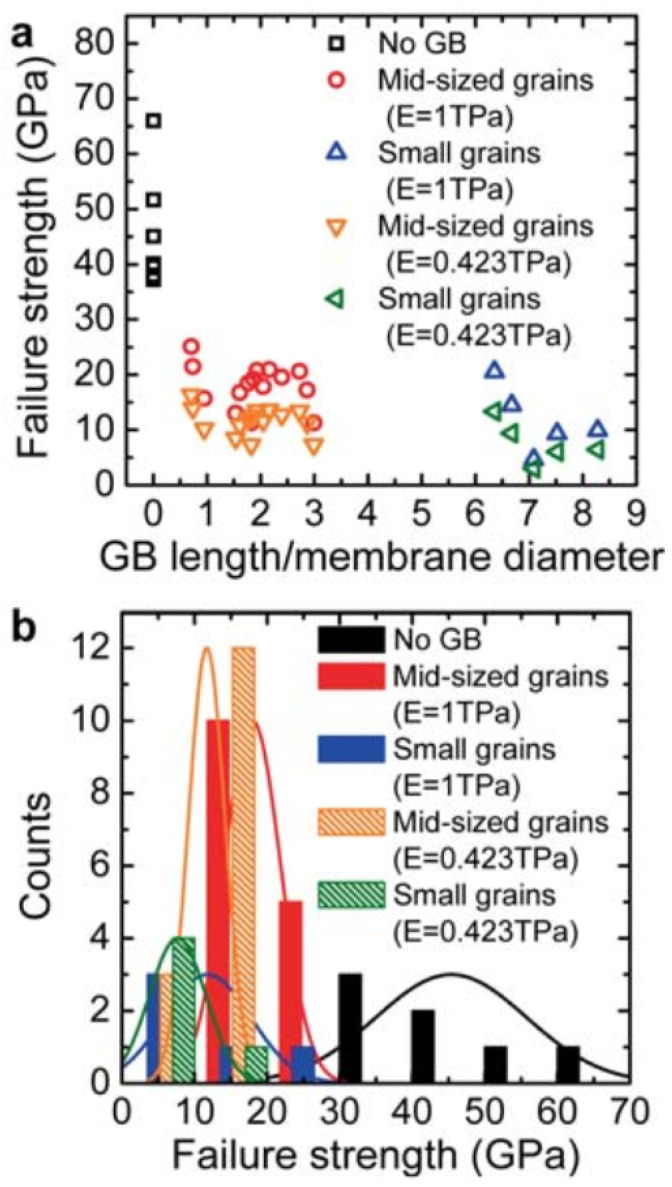
(**a**) Fracture Strength as a function of grain boundary size. (**b**) A distribution graph of number of samples with different grain sizes and fracture strengths [128]. Reprinted (adapted) with permission from [128]. Copyright ©2015, WILEY-VCH Verlag GmbH & Co. KGaA, Weinheim.

**Figure 7 micromachines-13-00027-f007:**
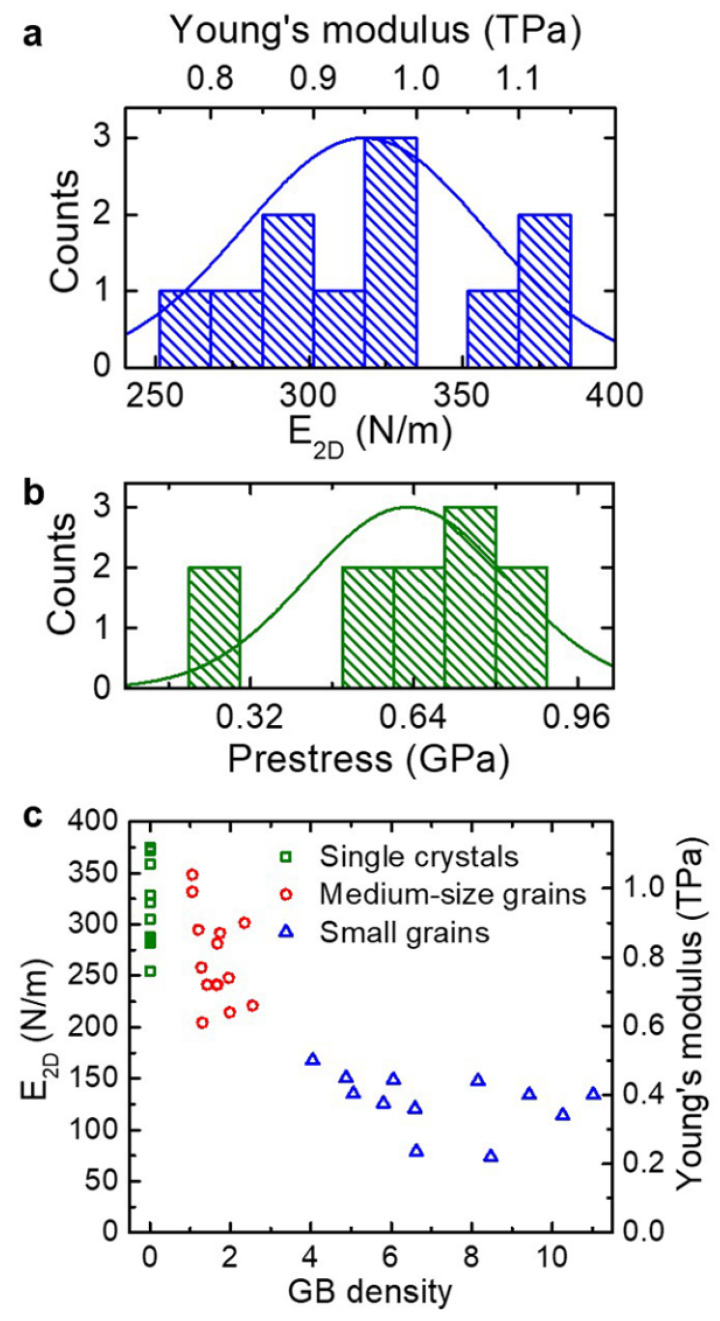
(**a**) Elastic Modulus as a function of grain boundary size. (**b**) A distribution graph of number of samples with different prestresses. (**c**) Elastic modulus as a function of grain boundary size [132]. Reprinted (adapted) with permission from [132]. Copyright ©2020, American Chemical Society.

**Figure 8 micromachines-13-00027-f008:**
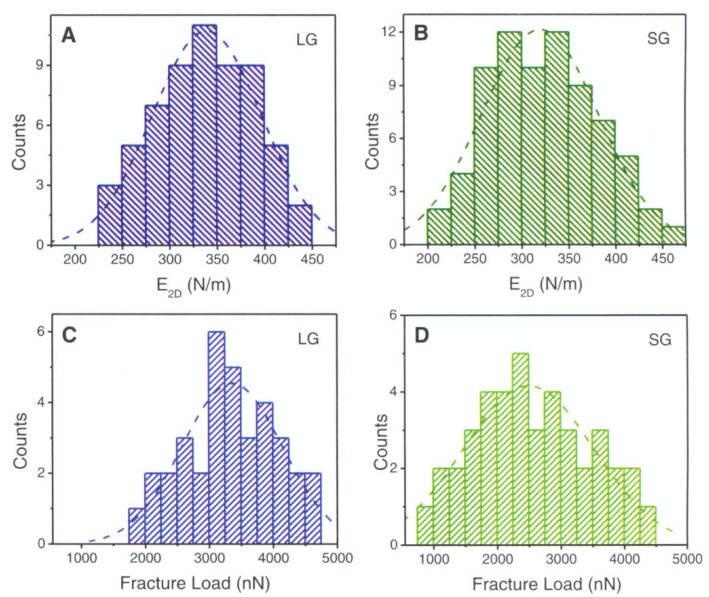
Comparison of modulus and fracture load for large grain (**A**,**C**) and small grain (**B**,**D**) single layer graphene [127]. Reprinted (adapted) with permission from [127]. Copyright ©2013, The American Association for the Advancement of Science.

**Figure 9 micromachines-13-00027-f009:**
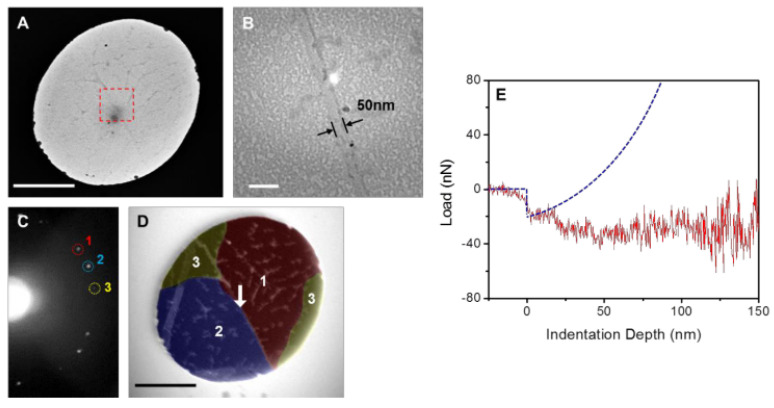
(**A**–**D**) Overlapping grain boundaries that appear as wrinkles under TEM; (**E**) No measurable force in this indentation experiment due to weak grain boundary [127]. Reprinted (adapted) with permission from [127]. Copyright ©2013, The American Association for the Advancement of Science.

**Figure 10 micromachines-13-00027-f010:**
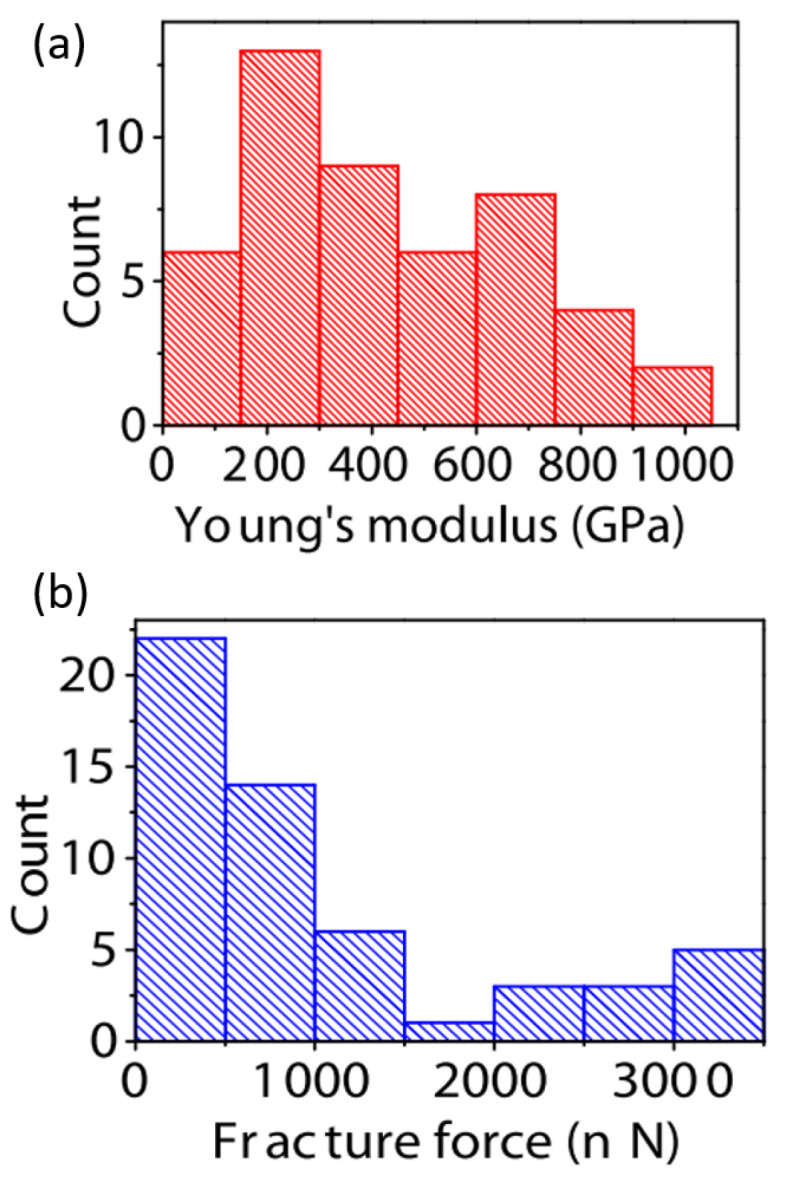
Measured Modulus (**a**) and Fracture force (**b**) of single layer graphene using nanoindentation with a bimodal distribution indicating overlapping and stitched grain boundaries [133]. Reprinted (adapted) with permission from [133]. Copyright ©2015, American Chemical Society.

**Figure 11 micromachines-13-00027-f011:**
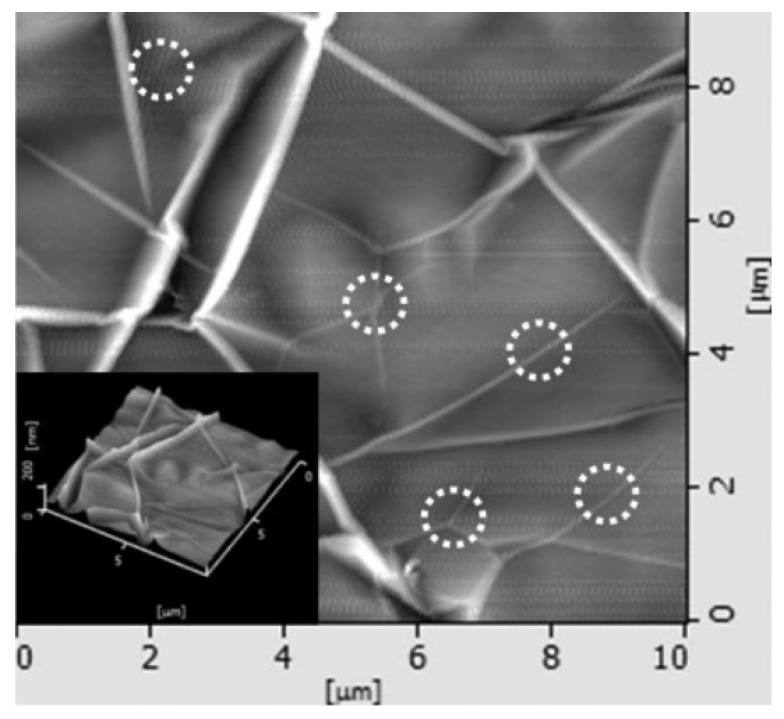
AFM image of graphene on a Ni substrate. Substrate grain boundary induced wrinkles are prominent throughout and smaller, thermal stress-induced wrinkles are circled [149]. Reprinted (adapted) with permission from [149]. Copyright ©2009, WILEY-VCH Verlag GmbH & Co. KGaA, Weinheim.

**Figure 12 micromachines-13-00027-f012:**
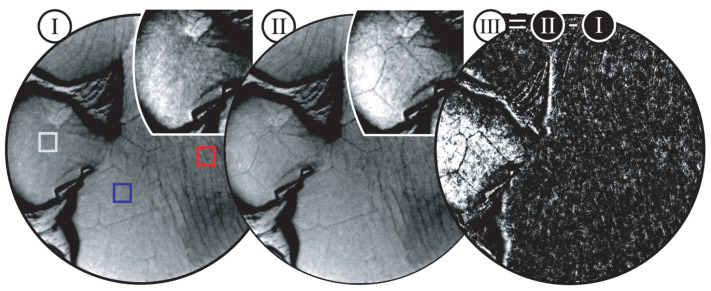
In-Situ LEEM image of graphene wrinkle formation on Ir(111). (field of view: 10 µm, electron energy: 2.5 eV) Wrinkles are formed on left side of image I and II. Change can be seen clearly in the difference image in III [151]. Reprinted (adapted) with permission from [151]. Copyright ©2009, Institute of Physics (the “Institute”) and IOP Publishing Limited.

**Figure 13 micromachines-13-00027-f013:**
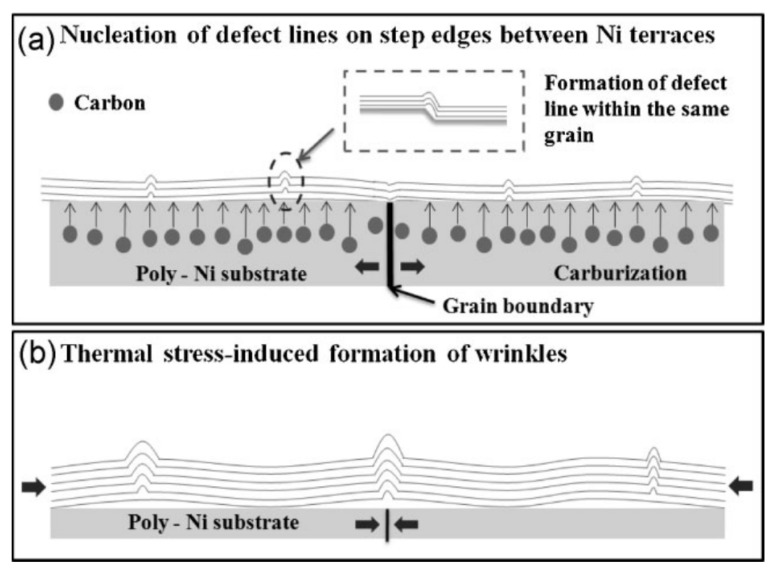
Schematic of wrinkle formation during CVD growth: (**a**) generation of wrinkles from nucleation of defect lines on step edges between Ni terraces (**b**) thermal-stress-induced formation of wrinkles around step edges and defect lines in multilayer graphene [149]. Reprinted (adapted) with permission from [149]. Copyright ©2009, WILEY-VCH Verlag GmbH & Co. KGaA, Weinheim.

**Figure 14 micromachines-13-00027-f014:**
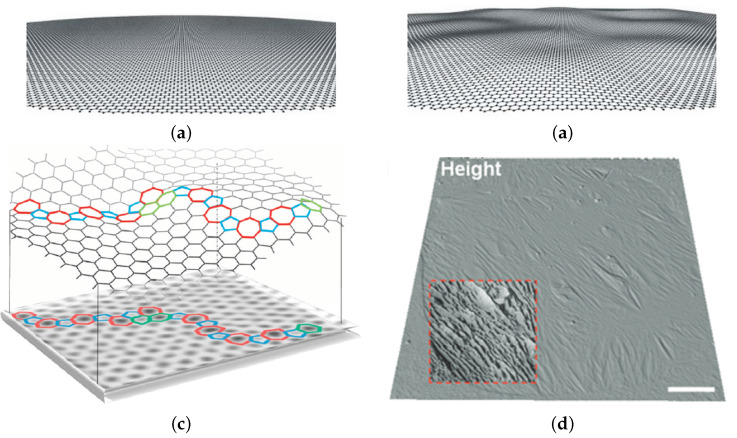
(**a**) Pristine suspended graphene [156] (**b**) Ripples in graphene caused by impurities [156]. Reprinted (adapted) with permission from [156]. Copyright ©2007, Nature Publishing Group. (**c**) Ripples in graphene caused by grain boundaries [142]. Reprinted (adapted) with permission from [142]. Copyright ©2011, American Chemical Society. (**d**) AFM image of ripples in suspended graphene [130]. Scale bar is 500 nm. Reprinted (adapted) with permission from [130]. Copyright ©2011, American Chemical Society.

**Figure 15 micromachines-13-00027-f015:**
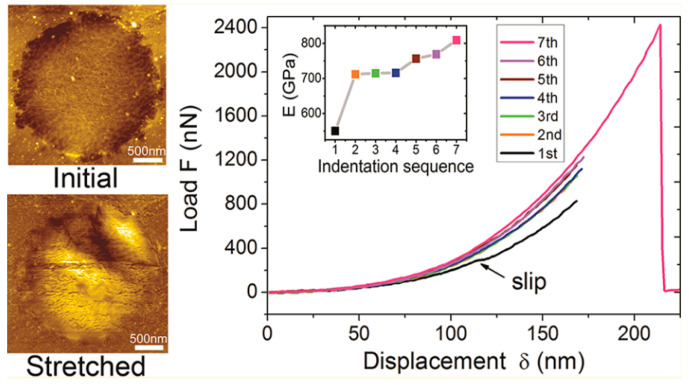
Stretch-induced stiffness enhancement [160]. Reprinted (adapted) with permission from [160]. Copyright ©2013, American Chemical Society.

**Figure 16 micromachines-13-00027-f016:**
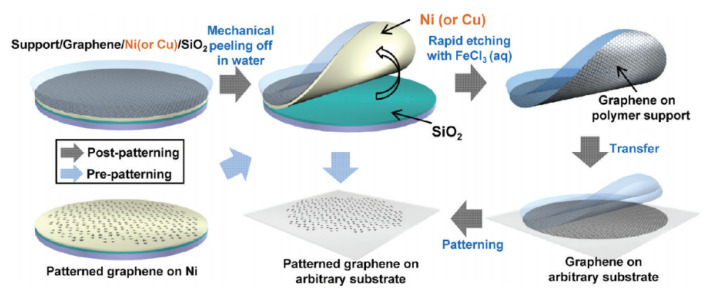
Illustration of the polymer-assisted wet-transfer technique. Graphene is grown on a Ni or Cu substrate, coated with a PMMA scaffold, the Ni or Cu layer is etched, and the graphene/PMMA is transferred to alternative substrate [165]. Reprinted (adapted) with permission from [165]. Copyright ©2010, American Chemical Society.

**Figure 17 micromachines-13-00027-f017:**
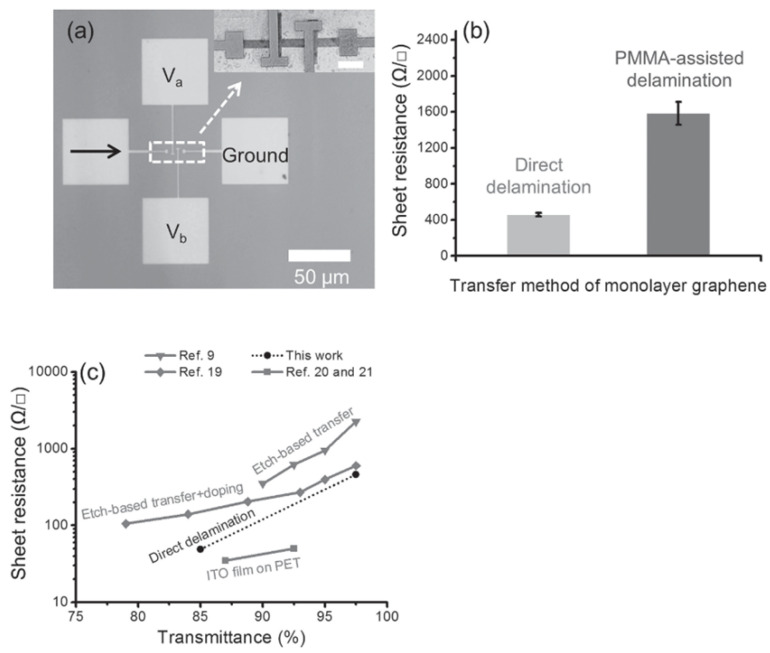
(**a**) Graphene with four point probe channel for sheet resistance measurement (**b**) Sheet resistance difference between direct delamination versus PMMA-assisted delamination (**c**) Sheet resistance from multiple transfer techniques [174]. Reprinted (adapted) with permission from [174]. Copyright ©2014, WILEY-VCH Verlag GmbH & Co. KGaA, Weinheim.

**Figure 18 micromachines-13-00027-f018:**
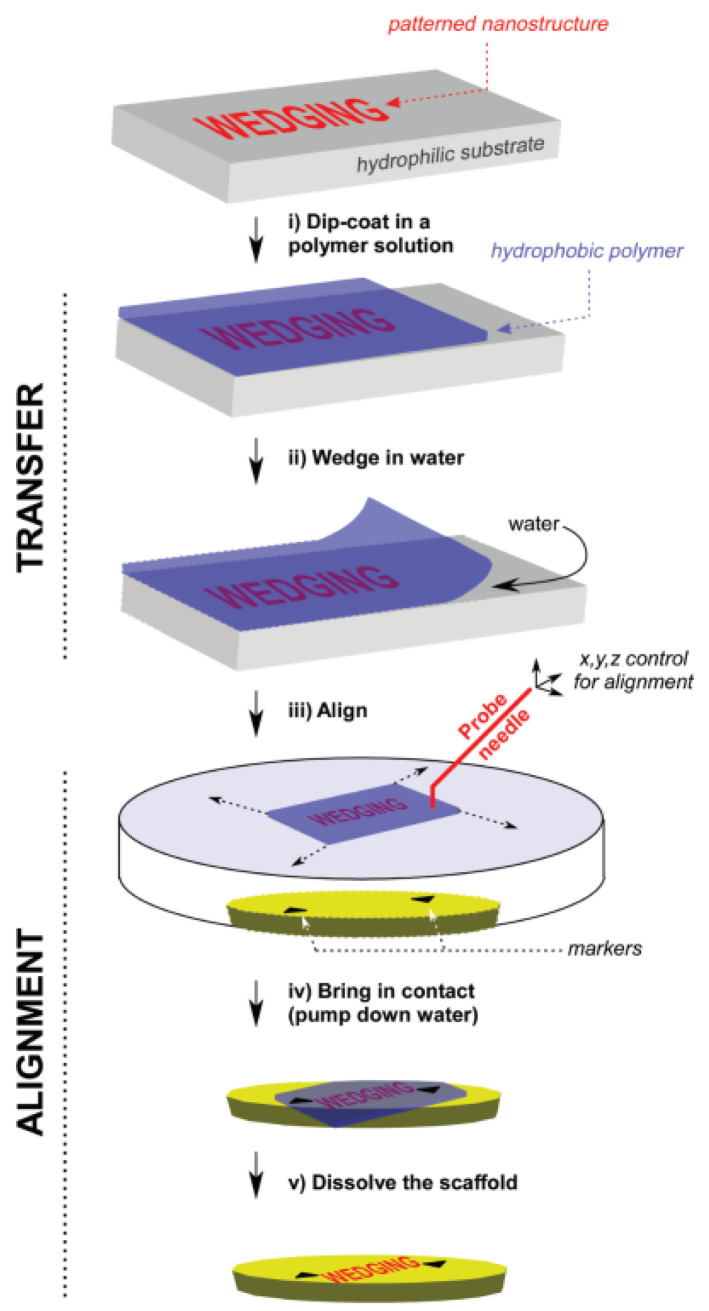
Graphene transfer using patterning, wedging transfer, and performing alignment in water [180]. Reprinted (adapted) with permission from [180]. Copyright ©2010, American Chemical Society.

**Figure 19 micromachines-13-00027-f019:**
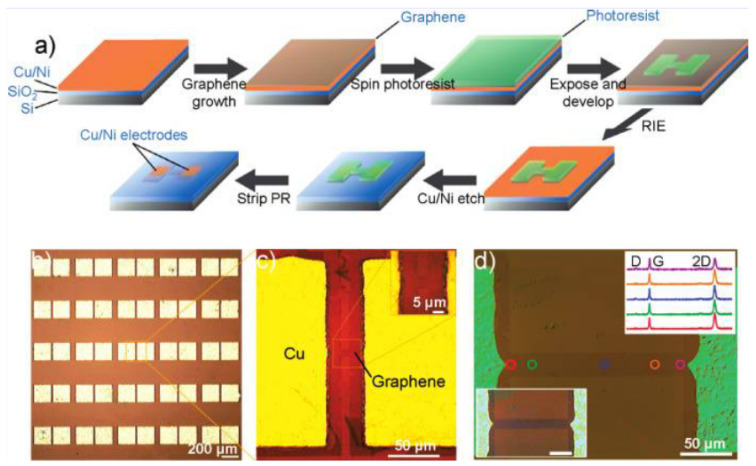
(**a**) A method to fabricate graphene transistor device without the use of transfer process in a wafer batch scale. (**b**) Optical image of substrate after batch fabrication. (**c**) Close view of graphene transistor consisting of two electrodes and suspended graphene in between. (**d**) Differential contrast image of a longer device with Raman spectroscopy measured on graphene [181]. Reprinted (adapted) with permission from [181]. Copyright ©2009, American Chemical Society.

**Figure 20 micromachines-13-00027-f020:**
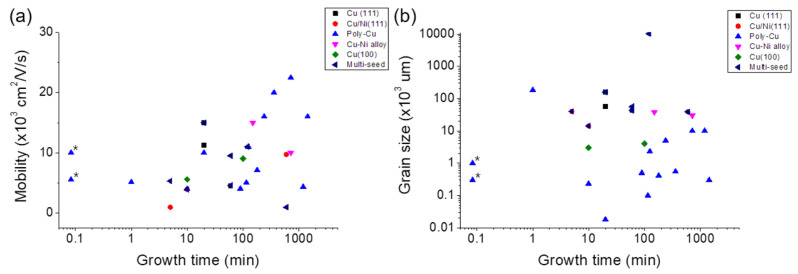
(**a**) Mobility by graphene growth time on different growth metal layer. (**b**) Grain sizes by graphene growth time on different growth metal layer [183,184,185]. Two data points with asterisks represent CVD growth of graphene by accelerated growth technique [186,187].

**Figure 21 micromachines-13-00027-f021:**
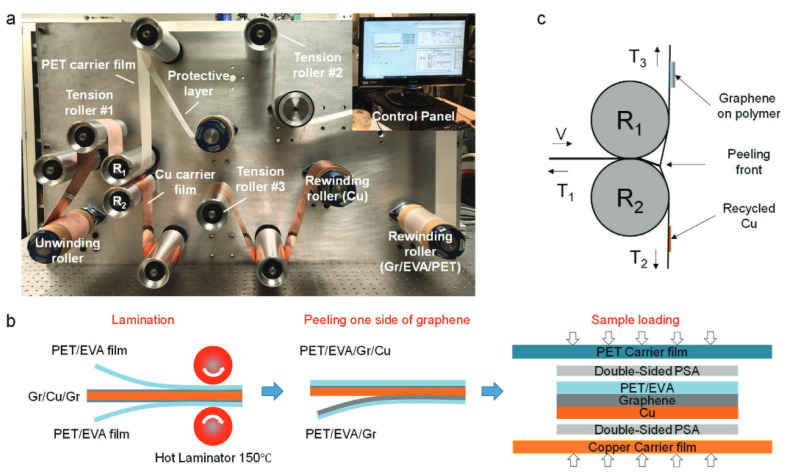
The process of roll-to-roll (R2R) graphene production on Cu foil. (**a**) Designed R2R graphene transfer system which separates PET/EVA/graphene and Cu foil during the process. (**b**) A schematic of how PET/EVA film is coated, peeled, and loaded to different carrier film. (**c**) A schematic of rollers peeling and separating polymer/graphene from Cu foil [177]. Reprinted (adapted) with permission from [177]. Copyright ©2021, Wiley-VCH GmbH.

**Figure 22 micromachines-13-00027-f022:**
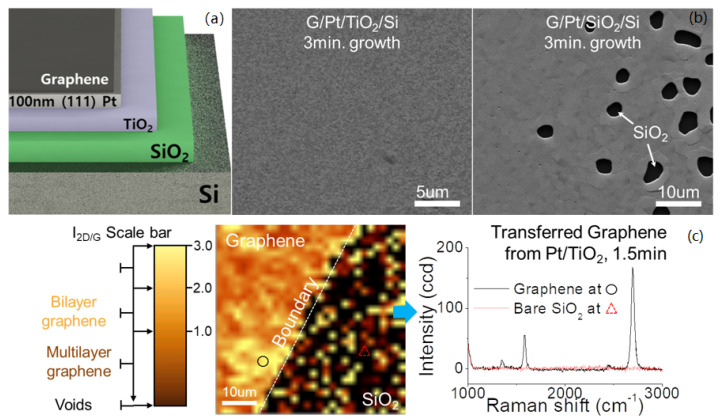
(**a**) Schematic of graphene grown on (111) Pt on TiO${}_{2}$/SiO${}_{2}$ substrate. (**b**) A SEM image showing less damage of graphene/(111) Pt on TiO${}_{2}$/SiO${}_{2}$/Si substrate by excessive dewetting in 1000 °C compared to (**c**) SEM image of damaged graphene/(111) Pt on SiO${}_{2}$/Si substrate [193]. Reprinted (adapted) with permission from [193]. Copyright ©2020, American Chemical Society.

**Figure 23 micromachines-13-00027-f023:**
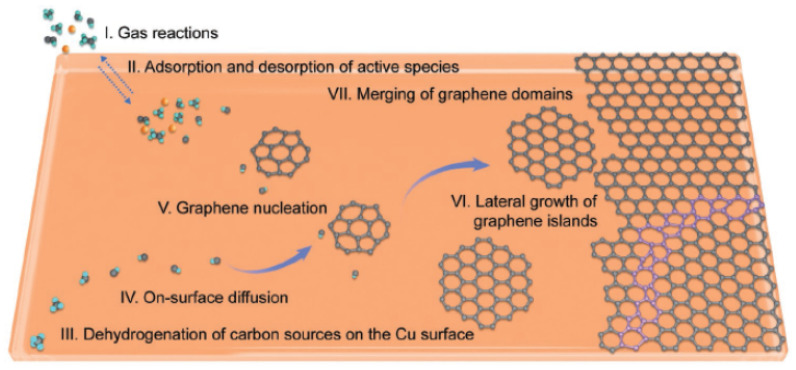
Two mechanisms of graphene growth, adsorption-desorption of carbon atoms and surface diffusion [209]. Reprinted (adapted) with permission from [209]. Copyright ©2020, WILEY-VCH Verlag GmbH & Co. KGaA, Weinheim.

**Figure 24 micromachines-13-00027-f024:**
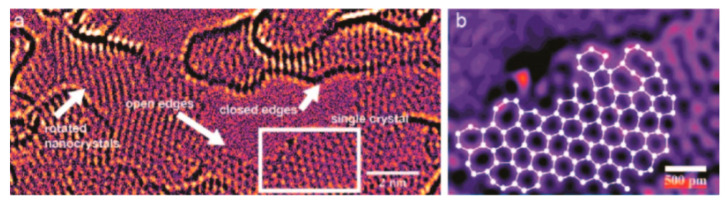
(**a**) Image of atomic structure measure in-situ AC-TEM at 700 °C (**b**) Close view with resolved graphene oxide structure [216]. Reprinted (adapted) with permission from [216]. Copyright ©2016, American Chemical Society.

**Figure 25 micromachines-13-00027-f025:**
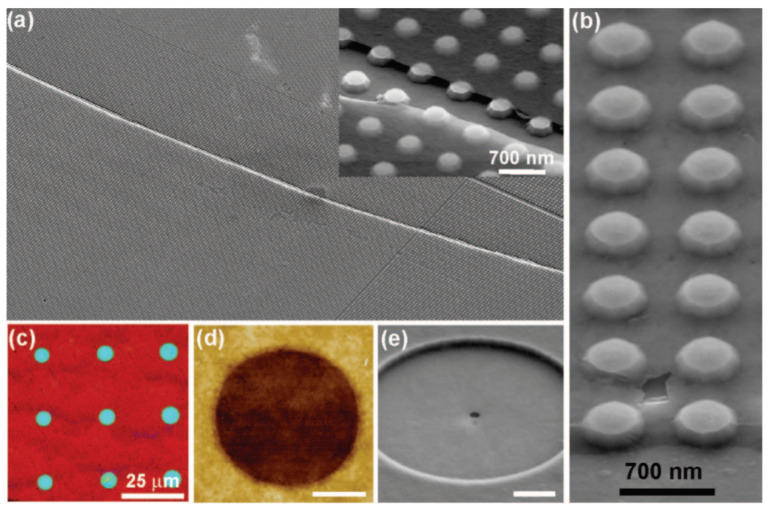
(**a**) A SEM image of suspended rGO resonator on patterened substrate and (**b**) close up view of rGO of 4nm thickness. (**c**) Optical image of rGO. (**d**) AFM height image of rGO and a drum. (**e**) A SEM image of a drum resonator after creating a hole [218]. Reprinted (adapted) with permission from [218]. Copyright ©2008, American Chemical Society.

**Table 1 micromachines-13-00027-t001:** Summary of reported Young’s Modulus and max strength of CVD and Exfoliated graphene.

Manufacturing Method	Number of Layers	E (GPa)	$\sigma_{\max}$ (GPa)	Ref.
Exfoliation	23–43	1000	-	[84]
Exfoliation	1	1000	130	[1]
Exfoliation	1	1026 ± 22	125 ± 0	[129]
Exfoliation	2	962 ± 24	107.7 ± 4.3	[129]
Exfoliation	3	980 ± 10	105.6 ± 6.0	[129]
Exfoliation	8	942 ± 3	85.3 ± 5.4	[129]
CVD on Cu	1	160	35	[130]
CVD on Cu (single grain)	1	-	90–94	[131]
CVD on Cu (poly-crystalline)	1	-	53–77	[131]
CVD on Cu	1	1000 ± 150	103–118	[127]
CVD on Cu (poly-crystalline small grain)	1	423–1000	11.8	[128]
CVD on Cu (poly-crystalline medium grain)	1	423–1000	18	[128]
CVD on Cu (single grain)	1	1000	45.4 ± 10.4	[128]
CVD on Cu (poly-crystalline small grain)	1	380 ± 80	-	[132]
CVD on Cu (poly-crystalline medium grain)	1	790 ± 130	-	[132]
CVD on Cu (single grain)	1	950 ± 120	-	[132]
CVD on Cu	1	423	28.7	[133]
CVD on Cu	2	435	31.5	[133]

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
