# Peer review of "Towards Repeatable, Scalable Graphene Integrated Micro-Nano Electromechanical Systems (MEMS/NEMS)"

_micromachines, 2021, doi:10.3390/mi13010027_

Round 1
Reviewer 1 Report
The present manuscript is a review on reproducible and scalable production of graphene MEMS/NEMS devices. I believe the topic is of interest to the field, since the scalable and reproducible integration of 2D materials in NEMS/MEMS devices remains a major challenge in the field. I therefore see the contents of the present manuscript as useful and timely. However, I believe the manuscript contains significant errors, and is missing useful or important citations (in particular concerning vibrating graphene devices), which should be addressed before this manuscript can be accepted for publication. I list these concerns below.
- The authors are not always clear in their figures whether what is shown concerns a MEMS device, or simply wrong about this. For example figure 16 in the captions says that this shows a MEMS/NEMS graphene device, but in the cited paper it is clearly mentioned that this graphene device is supported on a substrate. Actually, the paper in reference 164 concerns a transistor and it not a mechanical device at all.
A similar concern arises on page 4, line 137: here it is stated that graphene resonators are being used as charge sensors. However, the cited works are not graphene resonators. A better example of electrostatic charge detection with a graphene resonator can be found here: Chen, C., et al. Nature Phys 12, 240–244 (2016). - Section 3: I am missing a section about dynamical properties of graphene, which is important for many MEMS applications. See for example this review: Peter G Steeneken et al 2021 2D Mater. 8 042001This review also contains examples on how wrinkles/defects affect dynamical properties of graphene, for example mode shape variations or changes in the frequency dispersion. Currently, the manuscript only discusses static mechanical properties in detail.
- Section 4: Some examples are missing, such as graphene pressure sensors that operate by a change in resonance frequency. It would be useful to refer the reader to this review: Max C. Lemme, et al, Research, 2020, Article ID 8748602 for a comprehensive overview.
- Figure 1: The authors mention that the E2D is mostly the same between large grain, small grain and pristine graphene. However, within each dataset there is still considerable spread within this value. The consequence of this, is that if one builds two identical devices they may have completely different responsivities. This seems to be a recurring theme in a lot of figures that follow (fig. 4c, 5a,b). The authors should discuss this aspect in more detail, as too much attention it being given to the fracture strength.
- Page 6, line 122: in addition to reference 122, it would be good to cite: Zhang, Y., et al Nature 459, 820–823 (2009). The tunable bandgap is the most interesting property of bilayer graphene.
- Figure 6 and 7 and page 10: Regarding the overlapping boundaries. It is mentioned that this changes the elastic modulus by the grains slipping passed each other. However, I would expect this to be inelastic behaviour, so how can this affect the elastic coefficients?
- Page 17, line 449: The claim that hydrogen bubbles between growth substrate and graphene can damage the graphene needs to be supported with a citation.
- Figure 17: It is claimed that there is a correlation between growth time and mobility in Fig. 17(a). However, I do not see clear evidence in this figure. There are less datapoints with small growth times, making it less likely to see a large mobility by statistics alone.
- Section 6.2 Many of the transfer free processes described in this section are actually not used to make NEMS/MEMS devices, and the authors need to be clear on that. Another transfer free approach using Molybdenum catalyst to make suspended multilayer graphene devices it missing from this section, that has actually been used to make functioning sensors:
S. Vollebregt, et al 19th International Conference on Solid-State Sensors, Actuators and Microsystems (TRANSDUCERS), 2017, pp. 770-773
Joost Romijn et al 2021 Nanotechnology 32 335501 - Another aspect of scalability in graphene NEMS/MEMS that I am currently missing is characterization. Currently, the manuscript only shows AFM characterization which is impractical in high-volume production of graphene NEMS. Some research has been performed in order to try and scale up mechanical characterization, for example in these works:
Cartamil-Bueno et al, Nanoscale, 2017, 9, 7559-7564
Cartamil-Bueno et al, Nano Lett. 2016, 16, 11, 6792–6796 - The manuscript needs more careful proof-reading and spell-checking. I still found a lot of typo's in the work. Also the references need to be checked carefully, for example refs. 23 and 92 are double.
Author Response
Dear Reviewer 1,
We thank you for your contribution to review our paper and to provide valuable comments. We have carefully reviewed all of your comments and made edits. The response is quite long, therefore we have attached a pdf file.
Sincerely,
Joon Hyong Cho

Reviewer 2 Report
The review article of J. H. Cho et al. discusses the main merits of integrating graphene into MEMS/NEMS, current approaches for the mass production of graphene integrated devices, and propose solutions to overcome current manufacturing limits for the scalable and repeatable production of integrated devices based on graphene. The review treats very important topics and it is well organized and informative. The reviewer comments are below:
1) In fact, the pressure sensors based on graphene field-effect transistors have gained much attention in the last few years, where intriguing results have been recorded. In this context, I recommend to insert in the text body some sentences regarding this important point while citing the relevant papers, such as:
DOI: 10.1021/acs.nanolett.9b02978
DOI: 10.1016/j.aeue.2020.153346
DOI: 10.1038/ncomms14950
2) The field of NEMS/MEMS based on graphene seems in its infancy. For this reason, huge effort is directed to computational works including advanced modeling and simulations in order to predict the behaviors of the cutting-edge graphene based-NEMS/MEMS while paving the way to optimized fabrication and industry. In this context, I kindly recommend to add a section dealing with the computational works and emphasizing the main recorded findings.
2.1) Some relevant references should be inserted.
3) The treatment of some works dealing with the application of graphene-based MEMS/NEMS in medical and biomedical area can make this amazing review more informative.
4) In (4. Graphene in MEMS/NEMS) section, I recommend to insert a figure to show some advanced MEMS/NEMS switches, mass sensors, pressure sensors, …etc. This will be very beneficial and useful for the readers of micromachine journal. (Optional)
In my best opinion, I recommend the acceptance of this good review after a MINOR revision.
Good luck.
Author Response
Dear Reviewer 2,
We thank you for your contribution to review our paper and to provide valuable comments. We have carefully reviewed all of your comments and made edits. The response is quite long, therefore we have attached a pdf file.
Sincerely,
Joon Hyong Cho

Round 2
Reviewer 1 Report
The authors have sufficiently addressed my concerns, I especially like the additional figures 1-3 since they make it much more concrete what a "graphene MEMS/NEMS" looks like, which is nice for the reader that is not so familiar with these systems. I did notice that Reference 15 in the caption of figure 3 and page 5 is citing the wrong paper (should be: ACS Appl. Mater. Interfaces 2017, 9, 49, 43205–43210). The manuscript will be suitable for publication once this is addressed.